# AdCorDA: Classifier Refinement via Adversarial Correction and Domain Adaptation

## Abstract

This paper describes a simple yet effective technique for refining a pretrained classifier network. The proposed AdCorDA method consists of two stages - adversarial correction followed by domain adaptation. Adversarial correction uses adversarial attacks to correct misclassified training-set classifications. The incorrectly classified samples of the training set are removed and replaced with the adversarially corrected samples to form a new training set, and then, in the second stage, domain adaptation is performed back to the original training set. Extensive experimental validations show significant accuracy boosts of over 5% on the CIFAR-100 dataset and 1% on the CINIC-10 dataset. The technique can be straightforwardly applied to the refinement of weight-quantized neural networks, where experiments show substantial enhancement in performance over the baseline. The adversarial correction technique also results in enhanced robustness to adversarial attacks.

## 1 Introduction

Training of deep neural networks is an eternal struggle to improve accuracy, and many methods have been developed to eke out additional gains in performance from pretrained networks. These improvements are particularly important for smaller networks, such as those targeting edge devices, as their baseline performance is relatively low. In this paper, we present a novel alternative to standard neural network fine-tuning methods. We call this method AdCorDA, which stands for Adversarial Correction and Domain Adaptation. This method takes a classifier network pretrained using standard back-propagation methods and refines it with a domain adaptation step that adapts from a synthetic dataset, on which the pretrained network has perfect training accuracy, back to the original dataset. The synthetic dataset is constructed by performing adversarial correction on the dataset samples for which the pretrained network gets incorrect. Adversarial correction is the process of applying adversarial attacks to alter these dataset images such that the network classifies them correctly.

Our approach focuses on small networks to optimize performance on edge devices, and we therefore limit our experiments to small networks commonly used in such environments. Experiments show that our approach produces substantial enhancements in performance on image classification datasets, for both full-precision and quantized networks, and also induces a significant measure of robustness to adversarial attacks.

## 2 Background

As mentioned in the introduction, our approach has two stages - first modify the training set to increase the total training accuracy, and second adjust the weights to improve performance on the original training set. In our work we looked at two methods for altering the training set. The first method is based on *curriculum learning*, and the second is based on what we call *adversarial correction*.

### 2.1 Curriculum Learning

Curriculum learning, first proposed by Bengio et. al Bengio et al. (2009), aims to improve the speed and accuracy of network training, by presenting data samples from the training set in an ordered fashion. Typically, *easier* samples are presented before *difficult* samples as the training progresses. It is not obvious how to properly define the notions of "easy" and "hard", however, and indeed many

different definitions exist. Some of these definitions are based solely on the structure of the input examples, without consideration of the network being trained. Table 2 in the survey paper of Wang et. al Wang et al. (2021) lists no less than nineteen different types of pre-defined input difficulty measures that have been used to guide curriculum learning. But the difficulty of an input can also depend on the network being trained. Problems that some networks find difficult may be easy for other networks, and vice-versa. So-called *Self-Paced-Learning (SPL)* methods, such as proposed by Kumar et. al Kumar et al. (2010) use dynamic measures of problem difficulty that are provided by the network itself as it trains. In the SPL method, easy problems are defined as those problems for which the network's training loss is less than a (dynamically changing) threshold value. We propose to use the curriculum separation of the training set into easy and hard problems, as defined by the training loss threshold, for our approach. We consider that, over the original training set, our pre-trained network achieves a particular loss value. If we *remove* the training set samples for which the loss is above a threshold, then we are left with a (modified) training set for which the average (and maximum) loss is less than that of the original training set. To avoid having to set a suitable threshold value, we propose using the pre-trained network to define easy vs. hard using the simple expedient of considering easy problems to be ones the network classifies correctly. This will naturally result in a separation of input samples based on loss. We use this procedure to satisfy the first step of our AdCorDA process - altering the inputs to reduce the loss. Although we are not altering the loss for individual samples in this method, the average loss on a batch level *is* being altered.

## 2.2 Adversarial Correction

We can take the curriculum approach outlined in the previous section a step further, by doing what we call *adversarial correction* to further modify the training set. This results in a larger training set than the curriculum approach. The concept of adversarial attack is well known in the machine learning community Li et al. (2022). Given a classifier network trained on a particular dataset, an adversary can modify an input slightly in such a way that the network gives a different classification output. In this paper, rather than focusing on correct outputs being changed by adversarial attacks, we look at the effect of adversarial attacks on the outputs that the network already gets wrong. In such a situation, things cannot get any worse, as the network is already wrong, but they could get better if the adversarial perturbation of the input actually causes the network to provide the *correct* answer. We can help the process by using *targeted* attacks, where the target of the adversarial attack in this case is the correct output. But even non-targeted attacks may help by weakening support for the incorrect label relative to the true label. We will refer to this as *adversarial correction*, as opposed to *adversarial attack*.

Adversarial correction is well-suited to working with quantized networks, as some adversarial attacks do not need to compute the gradients with respect to the weights. However, many attacks do need gradient information and deep domain adaptation techniques generally require gradient-based optimization with respect to the weights to adapt models effectively across domains. Thus, in this paper, we focus on post-training quantization methods Jacob et al. (2018), and we apply the adversarial correction on the samples the quantized network gets wrong, rather than those of the full precision network.

## 2.3 Domain Adaptation

At this point in the method we have a modified dataset consisting of either only samples that the original network gets correct, or the same augmented with samples that have been adversarially corrected. Either way, our original trained network has an accuracy of $100\%$ on this modified dataset. But, how does this help us? After all, what we really want to do is increase accuracy (reduce the loss) on the original dataset, not some other dataset. This is the goal of the second stage of the input space training, namely finding a set of network weights that results in a lower loss on the original training set, starting from the modified training set.

Denote the original training set by $T$, and consider the altered training set $T'$ as our starting point for the second stage of the AdCorDA process. The original training set can be thought of as a distribution shift of the altered training set. How can we deal with this distribution shift, where we go from a distribution where the network does well (perfectly, in fact), to a distribution where the network performs less well? There is substantial literature addressing this problem: *domain adaptation*.

Domain adaptation methods aim to transfer knowledge about one domain (the *source* domain) into a second, similar, domain (the *target* domain) Zhang (2021). All domain adaptation methods have the goal of increasing performance on the target domain, starting from a network that does well on the source domain. Shen et. al Shen et al. (2023) showed that applying domain adaptation from easy to hard after the early stages of curriculum learning speeds up training. Motivated by these considerations we choose the final step in our AdCorDA method to be a domain adaptation from $T'$ to $T$.

# 3 METHODOLOGY: ADCORDA

## 3.1 OVERVIEW

Putting together the two stages of the input space training method as detailed above, we arrive at what we call the AdCorDA (Adversarial Correction and Domain Adaptation) method. The AdCorDA method proceeds as depicted in Fig. 1, with the following steps:

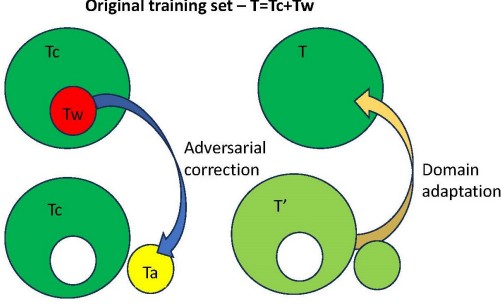

Figure 1: Overview of the proposed AdCorDA classifier refinement method. $T$ is the original training set; $T_c$ is the subset of $T$ that the pretrained network labels correctly, and $T_w$ the subset that is labeled incorrectly; $T_a$ is the set of samples that have been adversarially corrected; $T'$ is the union of $T_c$ and $T_a$. The network is adapted from $T'$ as the source domain back to $T$ as the target domain.

**Step 1:** Train a network to solve a classification problem using standard training techniques on a training set $T$.

**Step 2:** Separate the original set of training samples $T$ into two subsets: $T_c$ and $T_w$, where $T_c$ are training samples for which the trained network predicts correctly, and $T_w$ are training samples for which the network gives wrong predictions.

**Step 3:** For each sample in $T_w$, use adversarial attack techniques to create adversarial inputs, where in this case we wish to perturb the input such that the network gives the class provided by the training label (true label). Note that typically not all attacks will successfully coax the network to output the true label. Let the set of successfully perturbed samples be denoted as $T_a$. This may be smaller than the set $T_w$. This step can be omitted, in which case we are using the curriculum learning strategy. We refer to this in our experiments as the "None" or "Non-attack" case.

**Step 4:** Merge the subsets $T_c$ and $T_a$ into one new training set, $T'$. The samples for which the adversarial correction failed have been removed, so the accuracy of the network on $T'$ is 100%, and the number of elements in $T'$ may be less than that of the original dataset $T$.

**Step 5:** Seeing that $T$ and $T'$ represent two (overlapping) domains, do *domain adaptation* of the trained network, adapting from the corrected dataset $T'$ as the source domain, back to the original dataset $T$ as the target domain.

In the experiment section, we examine the effectiveness of the AdCorDA method, as well as an ablation case where we omit steps 3 and 4, using only the curriculum subset as $T'$.

## 3.2 Adversarial Attacks

To apply adversarial attacks on misclassified images of train domains, we use a selection of methods, including three major types of gradient-based attacks: basic iterative method Kurakin et al. (2017) and its variants, iterative least likely class Kurakin et al. (2017), decoupled direction and norm Rony et al. (2019), as well as a non-gradient-based salt and pepper noise attack. These are briefly described below.

**Untargeted Basic Iterative (BI)** Kurakin et al. (2017): This extends the "fast" method Goodfellow et al. (2015), which generates adversarial images through iterative processes using a small step size ($\alpha$) and clip pixel values of intermediate results at each step to ensure that they remain within an $\epsilon$-neighbourhood of the source image Kurakin et al. (2017):

$$\boldsymbol{X}_{N+1}^{\mathrm{BI}} = \underset{X,\epsilon}{\mathrm{Clip}}\Big\{\boldsymbol{X}_N^{\mathrm{BI}} + \alpha\,\mathrm{sign}\big(\nabla_X J(\boldsymbol{X}_N^{\mathrm{BI}}, y_{\mathrm{true}})\big)\Big\}, \quad \boldsymbol{X}_0^{\mathrm{BI}} = \boldsymbol{X}, \tag{1}$$

where $\boldsymbol{X}$ represents an image, $y_{\mathrm{true}}$ denotes the true class for the image $\boldsymbol{X}$, $J(\boldsymbol{X}, y)$ is the cross-entropy cost function of the neural network, $\underset{X,\epsilon}{\mathrm{Clip}}\{\boldsymbol{X'}\}$ is the per-pixel clipping function applied to the image $\boldsymbol{X'}$ to ensure it falls within an $L_\infty$ $\epsilon$-neighbourhood of the original image $\boldsymbol{X}$.

**Basic Iterative method with Highest probability class (BIH)**: When attacking a correct image, BI uses the true class gradient, where the highest-probability class aligns with the true class. However, when targeting an incorrect output, this changes – the highest probability class no longer represents the truth. Therefore, we adapt BI to use the gradient of the highest probability class to weaken the accuracy of the incorrect output, illustrated below:

$$\boldsymbol{X}_{N+1}^{\mathrm{BIH}} = \underset{X,\epsilon}{\mathrm{Clip}}\Big\{\boldsymbol{X}_N^{\mathrm{BIH}} + \alpha\,\mathrm{sign}\big(\nabla_X J(\boldsymbol{X}_N^{\mathrm{BIH}}, y_H)\big)\Big\}, \quad y_H = \underset{y}{\mathrm{argmax}}\{p(y|\boldsymbol{X})\}. \tag{2}$$

**Targeted Variant of Basic Iterative (VBI)**: In addition to the standard untargeted BI method, we created a targeted variant called VBI. Unlike BI (Eq. 1), which moves away from the true label, VBI (Eq. 3) operates in the opposite direction, moving towards the true label by negating the sign of the gradient sign function.

$$\boldsymbol{X}_{N+1}^{\mathrm{VBI}} = \underset{X,\epsilon}{\mathrm{Clip}}\Big\{\boldsymbol{X}_N^{\mathrm{VBI}} - \alpha\,\mathrm{sign}\big(\nabla_X J(\boldsymbol{X}_N^{\mathrm{VBI}}, y_{\mathrm{true}})\big)\Big\}. \tag{3}$$

**Iterative Least-Likely class (LL)** Kurakin et al. (2017): This method generates an attack targeting the least-likely class, as predicted by the trained model on the source image:

$$\boldsymbol{X}_{N+1}^{\mathrm{LL}} = \underset{X,\epsilon}{\mathrm{Clip}}\Big\{\boldsymbol{X}_N^{\mathrm{LL}} - \alpha\,\mathrm{sign}\big(\nabla_X J(\boldsymbol{X}_N^{\mathrm{LL}}, y_{\mathrm{LL}})\big)\Big\}, \quad y_{\mathrm{LL}} = \underset{y}{\mathrm{argmin}}\{p(y|\boldsymbol{X})\}. \tag{4}$$

The LL method moves the input in the direction of the gradient toward the least probable class. While this may lower the probability of the true class, it may also lower the probability of the maximum probability (incorrect) class by a larger amount, potentially correcting the output label.

**Decoupled Direction and Norm (DDN)** Rony et al. (2019): This attack is an iterative approach that refines the noise added to the input image in each iteration to make it adversarial. At iteration $i$, the adversarial input image, $x_i$, is generated as $x_i = x + \eta_i$, where $\eta_i$ is the noise with a norm of $\sigma_i$. If $x_i$ is adversarial, the norm of the next iteration noise is decreased i.e., $\sigma_{i+1} = \sigma_i(1 - \epsilon)$. Otherwise, the norm of the next noise is increased i.e., $\sigma_{i+1} = \sigma_i(1 + \epsilon)$. This process repeats until the minimum required perturbation is found Rony et al. (2019). The DDN method is a targeted attack that moves the network output towards the true label.

**Salt and Pepper noise (SP)**: A non-gradient-based attack that repeatedly adds SP noise to the input to fool the model.

To investigate the effect of our proposed method on the adversarial robustness of the corrected models, we evaluated the models against *AutoAttack* Croce & Hein (2020a) on CIFAR-10 and CIFAR-100 test sets. Composed of four different attacks from those used in our experiments for the correction, AutoAttack is a well-known, powerful, and diverse ensemble of parameter-free attacks. We applied the standard version of AutoAttack: APGD$_{\mathrm{CE}}$, targeted APGD$_{\mathrm{DLR}}$ Croce & Hein (2020a), targeted FAB Croce & Hein (2020b), and Square Attack Andriushchenko et al. (2020) with $\ell_\infty-$norm. The attacks were applied sequentially.

## 3.3 DOMAIN ADAPTATION

In the domain adaptation stage, we utilize Deep CORAL Sun & Saenko (2016), which aligns the second-order covariance matrices between a source domain and a target domain through CORAL loss. This alignment helps to bridge the distribution gap between the domains and improve the model's performance on the target domain. Aligning the implementation with the original paper, CORAL loss is only applied to the last classification layer in the neural networks. The total loss is the sum of the classification loss and the CORAL loss Sun & Saenko (2016), defined as

$$\mathcal{L}_{\text{loss}} = \mathcal{L}_{\text{class}} + \lambda \mathcal{L}_{\text{coral}}, \quad \mathcal{L}_{\text{coral}} = \frac{1}{4d^2}\|C_S - C_T\|_F^2, \tag{5}$$

where $\lambda$ is a weight between classification and CORAL loss, $C_S$ and $C_T$ are the covariance matrices of features induced by samples from the source domain and target domain, respectively, and the norm is the squared-matrix Frobenius norm. By minimizing the distance between the second-order statistics of the source and target domain feature representations, CORAL loss implicitly regularizes the learned feature space. In our application, the source domain is the adversarially corrected training dataset ($T'$) and the target domain is the original training dataset ($T$).

## 3.4 NETWORK QUANTIZATION

We also test the effectiveness of the AdCorDA method on network quantization, which reduces the precision of computations and weight storage by using lower bit-widths instead of floating-point precision. To obtain quantized models, we compress the baseline models using post-training static quantization (PTSQ) Jacob et al. (2018), which is one of the most common and fastest quantization techniques in practice. This technique determines the scales and zero points prior to inference. Specifically, we quantize the full-precision 32-bit (FP32) weights (e.g., $w \in [\alpha, \beta]$) and activations of the trained baseline models to 8-bit integer (INT8) values (e.g., $w_q \in [\alpha_q, \beta_q]$). The quantization process is defined as

$$w_q = \text{round}\left(\frac{1}{s}w + z\right), \quad s = \frac{\beta - \alpha}{\beta_q - \alpha_q}, \quad z = \text{round}\left(\frac{\beta\alpha_q - \alpha\beta_q}{\beta - \alpha}\right), \tag{6}$$

where $s$ is the scale and $z$ is the zero-point.

## 4 EXPERIMENTAL SETUP

*ResNet baseline models on CIFARs.* We validated our approach through experiments on the CIFAR-10 and -100 datasets, each containing 50K images, which are randomly split into 45K training data and 5K validation data. Each dataset has a separate test set of 10K images. We split the training and validation datasets using three random seeds: 1, 2, and 5. We first initialize ResNets He et al. (2016) of different sizes (i.e., ResNet-18, ResNet-34, ResNet-50) and EfficientNetV2-M Tan & Le (2021) with parameters pre-trained on the ImageNet dataset Deng et al. (2009) from PyTorch Paszke et al. (2019) and then fine-tune Yosinski et al. (2014) on the CIFAR training sets to obtain the corresponding baseline models. Input images are resized to $224 \times 224$ and use the same data transform. To determine the optimal hyper-parameters for our model, we perform a basic parameter grid search for the batch size, base learning rate, and weight decay of the stochastic gradient descent (SGD) optimizer. During the fine-tuning, we use an SGD optimizer Bottou (2010) with a momentum of 0.9, a weight decay of 1e-4, a batch size of 128 for ResNets on and of 64 (due to limitations in computing resources) for EfficientNetV2-M, a fixed learning rate of 1e-4, and we train for a total of 100 epochs on both CIFAR datasets. We define the fine-tuned models with the best validation accuracy as our baseline models.

*ResNet baseline models on CINIC-10* Darlow et al. (2018). We further validated our method on a larger dataset, CINIC-10. Constructed from ImageNet and CIFAR-10, it allocates 90K images for training, validation, and testing, respectively. To mitigate potential pre-trained model exposure to the training data, we utilize an alternative pre-trained model trained on a separate large-scale dataset, diverging from ImageNet, for unbiased fine-tuning. We first initialize ResNets with parameters pre-trained on the Places365-Standard dataset Zhou et al. (2017), which train set contains ~1.8M images from 365 scene categories and each category has at most 5K images. Then, we shuffle

the dataset and fine-tune the CINIC-10 training sets to get its baseline models using the same data transform and preprocessing as the pre-trained Places365 models, including the implementation of random crop functions for better model generalization. Ensuring reproducibility in the dataloader is imperative for subsequent adversarial correction steps. For fine-tuning on CINIC-10, we use a batch size of 64 and a fixed learning rate of 1e-3.

*Adversarial attack experiments on CIFARs and CINIC-10.* We apply adversarial attacks on misclassified training images while the model is in evaluation mode. For the DDN and SP attacks, we use the default hyper-parameters provided by the Foolbox framework Rauber et al. (2017; 2020). Note that the input images are subject to the ImageNet transformation with a lower and upper bound of 0 and 1, respectively. The BI and LL attacks are applied according to the experimental setting outlined in Kurakin et al. (2017).

*AutoAttack experiments on CIFAR.* We set $\epsilon$ to 5e-4 for all of the AutoAttack experiments. Other AutoAttack parameters, such as iterations and number of restarts, are identical to the parameters used in the standard version. The batch size used for ResNet-18, ResNet-34, and EfficientNetV2-M experiments is 512, 512, and 100, respectively. We reported the average test accuracy obtained across three random seeds.

*Domain adaptation experiments on CIFARs and CINIC-10.* Our Deep CORAL experimental setup follows the guidelines in Sun & Saenko (2016). However, we deviate by using batch sizes of 16 for ResNets on CIFAR-10 and CIFAR-100, of 16/32 for EfficientNetV2-M on CIFAR-10/100, and 64 for CINIC-10, differing from the original paper's settings. Also, we use a fixed learning rate of 1e-3 on CIFARs and 1e-4 on CINIC-10. We set $\lambda$ as 1/750 for CIFAR-10, 1/25 for CIFAR-100, and 1/2 for CINIC-10. We initialize the DA model with weights from the baseline models rather than using the pre-trained models, then apply 20-epoch DA training. These adjustments ensure a fair comparison with baseline models. When applying DA to quantized models, we facilitate the back-propagation process by approximating the gradients in these models. We achieve this approximation by utilizing the gradients derived from their corresponding full-precision models. This approach enables us to effectively conduct back-propagation on the quantized models. We define the best adapted model as the one that achieves the highest validation accuracy on the target domain, the original dataset $T$.

*Post-training static quantization.* We apply PTSQ on baseline models using the built-in quantization modules provided by PyTorch. These modules facilitate the fusion of different model components, calibration of the model using training data to determine suitable scale factors, and the actual quantization of weights and activations in the model. Note that we perform the adversarial correction on the training samples that the quantized network gets wrong, rather than the full precision network.

## 5 RESULTS AND DISCUSSION

Our approach improves the baseline performance through two steps: adversarial correction and domain adaptation. The "none" case involves only domain adaptation, providing most of the performance improvement, while the adversarial correction provides incremental improvement. The "*none*" attack case corresponds to the situation where we do not apply any adversarial correction, effectively relying only on the curriculum modification of the training set. Instead of training from an easy to a hard curriculum, we apply domain adaptation to go from easy to hard curriculum. Inspired by curriculum learning, we consider data in different difficulty levels as data with different distributions, i.e., in distinct domains. Therefore, instead of training on more difficult samples, we can transfer knowledge from one domain of the dataset (e.g., source domain - an easy domain with 100% accuracy) to a related but different domain (e.g., target domain - hard domain) within the dataset. This is inspired by the work in Shen et al. (2023), who used domain adaptation in this way in a standard curriculum learning process. They found that this form of curriculum learning was much faster than standard curriculum learning. Our approach differs in two significant ways from method stated in Shen et al. (2023): first, it does not require an external scoring function to create the easy/hard curriculum, instead using the ground truth fidelity. Second, we enhance the source domain by adding the adversarial corrected data samples, thereby improving the domain adaptation. One could argue that in doing adversarial correction we are performing a type of dataset augmentation, by creating new samples with known labels. However, we are not training on this augmented dataset in a standard manner. Instead, the removal of the incorrect samples and the addition of the corrected samples provides a more pure representation of the domain that the initial network does well on,

thereby enhancing the effectiveness of the subsequent domain adaptation step. Indeed, even just removing the incorrect samples, without adding the adversarial corrections, provides a significant benefit to the domain adaptation step.

## 5.1 ADVERSARIAL CORRECTION OF FULL PRECISION MODELS

The training, validation, and test accuracy of various networks obtained by applying AdCorDA for different attack methods on CIFAR-10 and CIFAR-100 are presented in Tab. 1. Our approach overall enhances the model performance by as much as 2.64% and 5.23% on CIFAR-10 and CIFAR-100, respectively, when utilizing ResNets of various sizes. As for the effect of our method when applied to EfficientNet, we note an enhancement ranging from approximately 0.7% to 1.1% across CIFAR datasets. More specifically, the ResNet-34 baseline model, operating at full precision, achieved a test accuracy of 78.41% on CIFAR-100. Our adversarial correction method, using a DDN attack, improves the test accuracy to 83.64%, representing a notable increase of 5.23%. In addition, we applied AdCorDA to the larger CINIC-10 dataset, and the performance of our pipeline is presented in Tab. 2. Our approach resulted in approximately a 1% improvement in ResNet model performance.

Table 1: Accuracy (%) of FP32 baselines (denoted as BL), which is fine-tuned on the CIFAR train domains, and accuracy of baselines after applying our approach (denoted as BL-IST) using different attacks to generate adversarial domains. "Corr." represents correction rates after adversarial attacks.

| Model | Approach | Attack | CIFAR-10 | | | CIFAR-100 | | |
|---|---|---|---|---|---|---|---|---|
| | | | Corr. Rate | Test | $\Delta$ Acc | Corr. Rate | Test | $\Delta$ Acc |
| ResNet-18 (11.19M) | BL | - | - | $93.29_{\pm 0.37}$ | - | - | $77.04_{\pm 0.08}$ | - |
| | BL-IST | None | - | $95.57_{\pm 0.13}$ | +2.28 | - | $80.27_{\pm 0.74}$ | +3.23 |
| | BL-IST | LL | 55/176 | $\mathbf{95.93}_{\pm 0.15}$ | $\mathbf{+2.64}$ | 70/451 | $80.93_{\pm 0.46}$ | +3.90 |
| | BL-IST | BIH | 99/176 | $95.87_{\pm 0.24}$ | +2.58 | 51/451 | $\mathbf{80.99}_{\pm 0.45}$ | $\mathbf{+3.96}$ |
| | BL-IST | VBI | 175/176 | $95.77_{\pm 0.06}$ | +2.48 | 446/451 | $80.54_{\pm 0.80}$ | +3.50 |
| | BL-IST | DDN | 176/176 | $95.84_{\pm 0.07}$ | +2.55 | 451/451 | $80.82_{\pm 0.35}$ | +3.79 |
| | BL-IST | SP | 45/176 | $95.80_{\pm 0.08}$ | +2.51 | 43/451 | $80.89_{\pm 0.61}$ | +3.86 |
| ResNet-34 (21.30M) | BL | - | - | $94.22_{\pm 0.06}$ | - | - | $78.41_{\pm 0.10}$ | - |
| | BL-IST | None | - | $96.40_{\pm 0.05}$ | +2.18 | - | $82.98_{\pm 0.07}$ | +4.57 |
| | BL-IST | LL | 25/80 | $96.31_{\pm 0.12}$ | +2.09 | 370/2538 | $82.69_{\pm 0.12}$ | +4.28 |
| | BL-IST | BIH | 46/80 | $96.36_{\pm 0.07}$ | +2.14 | 655/2538 | $83.31_{\pm 0.06}$ | +4.90 |
| | BL-IST | VBI | 80/80 | $96.26_{\pm 0.12}$ | +2.04 | 2490/2538 | $83.26_{\pm 0.45}$ | +4.85 |
| | BL-IST | DDN | 80/80 | $\mathbf{96.71}_{\pm 0.05}$ | $\mathbf{+2.49}$ | 2538/2538 | $\mathbf{83.64}_{\pm 0.06}$ | $\mathbf{+5.23}$ |
| | BL-IST | SP | 23/80 | $96.22_{\pm 0.05}$ | +2.00 | 118/2538 | $83.25_{\pm 0.29}$ | +4.84 |
| ResNet-50 (23.57M) | BL | - | - | $94.32_{\pm 0.59}$ | - | - | $79.74_{\pm 0.19}$ | - |
| | BL-IST | None | - | $\mathbf{96.61}_{\pm 0.12}$ | $\mathbf{+2.29}$ | - | $\mathbf{83.89}_{\pm 0.22}$ | $\mathbf{+4.15}$ |
| | BL-IST | LL | 46/131 | $96.31_{\pm 0.11}$ | +1.99 | 60/775 | $83.11_{\pm 0.48}$ | +3.37 |
| | BL-IST | BIH | 69/141 | $96.11_{\pm 0.16}$ | +1.79 | 261/775 | $83.03_{\pm 0.43}$ | +3.29 |
| | BL-IST | VBI | 130/131 | $96.50_{\pm 0.18}$ | +2.18 | 741/775 | $82.87_{\pm 0.07}$ | +3.13 |
| | BL-IST | DDN | 131/131 | $96.35_{\pm 0.12}$ | +2.03 | 775/775 | $83.03_{\pm 0.07}$ | +3.29 |
| | BL-IST | SP | 17/131 | $96.30_{\pm 0.15}$ | +1.98 | 45/775 | $83.25_{\pm 0.32}$ | +3.51 |
| EfficientNetV2-M (52.99M) | BL | - | - | $97.15_{\pm 0.14}$ | - | - | $86.88_{\pm 0.46}$ | - |
| | BL-IST | None | - | $97.76_{\pm 0.14}$ | +0.61 | - | $87.36_{\pm 0.57}$ | +0.48 |
| | BL-IST | LL | 3/9 | $97.82_{\pm 0.08}$ | +0.67 | 17/54 | $87.52_{\pm 0.45}$ | +0.64 |
| | BL-IST | BIH | 6/9 | $97.82_{\pm 0.09}$ | +0.68 | 23/54 | $\mathbf{88.00}_{\pm 0.10}$ | $\mathbf{+1.12}$ |
| | BL-IST | VBI | 8/9 | $97.80_{\pm 0.04}$ | +0.65 | 46/54 | $87.76_{\pm 0.16}$ | +0.88 |
| | BL-IST | DDN | 9/9 | $\mathbf{97.86}_{\pm 0.06}$ | $\mathbf{+0.71}$ | 54/54 | $87.81_{\pm 0.10}$ | +0.93 |
| | BL-IST | SP | 4/9 | $97.70_{\pm 0.12}$ | +0.55 | 18/54 | $87.89_{\pm 0.19}$ | +1.01 |

Table 2: Accuracy (%) of baseline models after applying the AdCorDA approach on CINIC-10.

| Model | Approach | Attack | Corr. Rate | Train | Test |
|---|---|---|---|---|---|
| ResNet-18 | BL | - | - | 94.75 | 84.09 |
| | BL-IST | None | - | 94.47 | 84.88 (+0.79) |
| | BL-IST | DDN | 4723/4723 | 94.97 | 84.99 (**+0.90**) |
| ResNet-50 | BL | - | - | 95.98 | 86.60 |
| | BL-IST | None | - | 95.35 | 87.60 (**+1.00**) |
| | BL-IST | DDN | 3620/3620 | 95.77 | 87.58 (+0.98) |

Upon incorporating adversarial correction using the LL attack on the training set, we observed a decrease in the initial training loss from 0.254 (on the original training set $T$) to 0.173 (on the

corrected training set $T'$) on CIFAR-100. This shows that the adversarial correction does indeed reduce the training loss.

## 5.2 Adversarial Correction of Quantized Models

Table 3 shows that our method also improves the baseline performance of quantized networks. For example, the full precision baseline ResNet-34 achieves a test accuracy of 78.41% on CIFAR-100. The Int8 quantized baseline ResNet-34 has a test accuracy of 77.13% on CIFAR-100. When applying our method using the BIH attack on quantized ResNet-34, it achieves a test accuracy of 82.18% - an improvement of +5.05% over its original quantized network (and an improvement of +3.77% over its original full precision network!).

The quantized ResNet-34 network after using our adversarial correction technique achieves a higher accuracy (82.18% on CIFAR-100) than even that of a normally trained full-precision ResNet-152 baseline model (81.52%), while significantly reducing the model size (20.76MB vs 223.49MB).

Table 3: Accuracy (%) of quantized (Int8) ResNets of various sizes obtained after applying PTSQ on its baseline, and the accuracy of Int8 ResNets using our approach.

| Model | Approach | Attack | CIFAR-10 | CIFAR-100 |
|---|---|---|---|---|
| ResNet-18 | BL | - | $93.29_{\pm0.37}$ | $77.04_{\pm0.08}$ |
| | PTSQ | - | $92.42_{\pm0.17}$ | $76.06_{\pm0.94}$ |
| | PTSQ-IST | None | $95.18_{\pm0.09}$ (+2.76) | $79.15_{\pm0.26}$ (+3.09) |
| | PTSQ-IST | BIH | $\mathbf{95.48}_{\pm0.18}$ (**+3.07**) | $79.53_{\pm0.58}$ (+3.47) |
| | PTSQ-IST | SP | $95.29_{\pm0.06}$ (+2.93) | $\mathbf{79.79}_{\pm0.49}$ (**+3.73**) |
| ResNet-34 | BL | - | $94.22_{\pm0.06}$ | $78.41_{\pm0.10}$ |
| | PTSQ | - | $93.36_{\pm0.09}$ | $77.13_{\pm0.45}$ |
| | PTSQ-IST | None | $\mathbf{96.08}_{\pm0.20}$ (**+2.72**) | $81.94_{\pm0.45}$ (+4.81) |
| | PTSQ-IST | BIH | $96.05_{\pm0.07}$ (+2.69) | $\mathbf{82.18}_{\pm0.20}$ (**+5.05**) |
| | PTSQ-IST | SP | $95.83_{\pm0.19}$ (+2.47) | $82.12_{\pm0.20}$ (+4.99) |

## 5.3 Early Stopping for Adversarial Correction

We investigate the effect of varying the number of baseline training epochs on the overall performance of our pipeline, as depicted in Fig. 2. Instead of fine-tuning pre-trained models for 100 epochs to build baselines, we conduct fine-tuning for fewer epochs, such as 20 or 40 epochs. The corresponding baseline performance with different numbers of training epochs is denoted as "BL" in Fig. 2a. Subsequently, we apply adversarial correction to the misclassified samples from each baseline, with the performance of both the none case (0% correction rate) and DDN case (100% correction rate) presented in Fig. 2a. The total number of misclassified samples increases as the training accuracy of the baselines decreases due to early stopping. Our findings reveal that with only 20 epochs of baseline training, our approach demonstrates a significant improvement (from 73.48% to 80.57%) through direct DA (i.e., none case) and achieves further enhancement (from 80.57% to 83.51%) with adversarial correction. As the number of altered samples decreases due to higher baseline performance, the effect of adversarial correction diminishes. Nevertheless, DA consistently contributes to improvements in the baselines. Moreover, as shown in Fig. 2b, we show the effect of doing DDN adversarial correction using various correction rates on BL. We see that as the number of corrected incorrect samples increases, we obtain nearly linear improvements in accuracy gain.

## 5.4 Enhanced Robustness to Adversarial Attacks

Our adversarial correction technique has many similarities to *adversarial training* methods for enhancing robustness to adversarial attacks. Such methods generate adversarial examples, for which networks give the wrong answer, and add these as augmentations of the original dataset. Fine-tuning on the augmented dataset leads to enhanced robustness against adversarial attacks Madry et al. (2017). Our approach is similar in that we create new images resulting from adversarial attacks, and use these in concert with images from the original dataset in further training. There are significant differences, however, between our method and standard adversarial training. First, we do not augment the original dataset, but instead replace some of the samples in the original dataset with the adversarial examples. Second, the adversarial attacks are only applied to samples that the network gets wrong, rather than

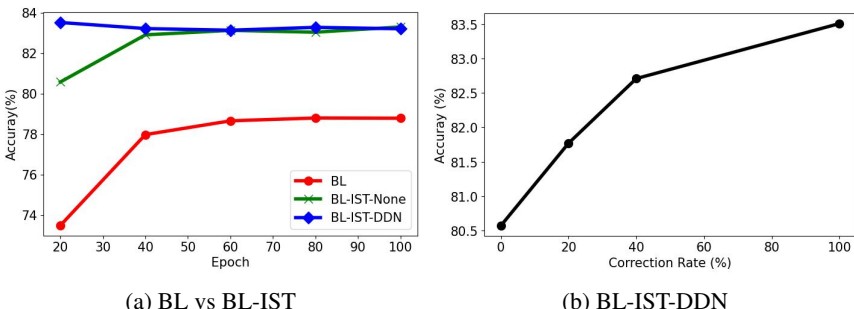

(a) BL vs BL-IST          (b) BL-IST-DDN

Figure 2: Performance comparison of ResNet-34 on CIFAR-100. (a) our pipeline results with BL achieved through fine-tuning across various epoch counts; (b) our pipeline results with DDN adversarial correction using different correction rates on BL obtained by fine-tuning with 20 epochs.

samples that the network gets right, and we only keep the adversarial examples which are corrective - that the network now gets right. Finally, rather than doing fine-tuning using standard training on the augmented training set, we do domain adaptation from the adversarially corrected training set to the original training set. We tested the robustness of ResNets to the AutoAttack suite of attacks Croce & Hein (2020a). As seen in Tab. 4, our method provides significant robustness to adversarial attacks. For CIFAR-10 with ResNet-18 we see an improvement from 15.92% on the baseline model to 50.97% on the adversarially corrected model with the SP correction method. On CIFAR-100 with ResNet-18 we see an improvement from 7.56% to 21.8%. Note that using only curriculum domain adaptation (the "None" case) also gives significant robustness. While current state-of-the-art robust network techniques get higher accuracies under attack than ours (e.g., 27.67% on CIFAR-100 by Addepalli et al. (2022) and 55.54% on CIFAR-10 by Sehwag et al. (2021), both with ResNet-18), our focus is on attaining higher clean (before attacks) accuracies, and the enhanced robustness is a welcome byproduct. Jointly optimizing both clean accuracy and adversarial robustness is an interesting avenue for future work.

Table 4: Accuracy (%) of FP32 baselines and adapted models using our approach on the clean and adversarially perturbed CIFAR test sets. AutoAttack is used to generate the adversarial samples.

| Model | Approach | Attack | CIFAR-10 | | CIFAR-100 | |
|---|---|---|---|---|---|---|
| | | | Clean | AutoAttack | Clean | AutoAttack |
| ResNet-18 | BL | - | $93.29_{\pm 0.37}$ | $15.92_{\pm 1.67}$ | $77.04_{\pm 0.08}$ | $7.56_{\pm 1.14}$ |
| | BL-IST | None | $95.57_{\pm 0.13}$ | $47.63_{\pm 1.74}$ | $80.27_{\pm 0.74}$ | $20.65_{\pm 0.82}$ |
| | BL-IST | DDN | $\mathbf{95.84}_{\pm 0.07}$ | $47.97_{\pm 0.10}$ | $80.82_{\pm 0.35}$ | $21.66_{\pm 0.94}$ |
| | BL-IST | SP | $95.80_{\pm 0.08}$ | $\mathbf{50.97}_{\pm 0.72}$ | $\mathbf{80.89}_{\pm 0.61}$ | $\mathbf{21.80}_{\pm 1.87}$ |
| ResNet-34 | BL | - | $94.22_{\pm 0.06}$ | $13.80_{\pm 1.05}$ | $78.41_{\pm 0.10}$ | $7.90_{\pm 0.62}$ |
| | BL-IST | None | $96.40_{\pm 0.05}$ | $50.54_{\pm 2.90}$ | $82.98_{\pm 0.07}$ | $22.37_{\pm 1.39}$ |
| | BL-IST | DDN | $\mathbf{96.71}_{\pm 0.05}$ | $51.03_{\pm 2.89}$ | $\mathbf{83.64}_{\pm 0.06}$ | $20.68_{\pm 2.19}$ |
| | BL-IST | SP | $96.22_{\pm 0.05}$ | $50.13_{\pm 2.41}$ | $83.25_{\pm 0.29}$ | $\mathbf{24.47}_{\pm 0.31}$ |
| EfficientNetV2-M | BL | - | $97.15_{\pm 0.14}$ | $15.07_{\pm 0.78}$ | $86.88_{\pm 0.46}$ | $11.16_{\pm 0.45}$ |
| | BL-IST | None | $97.76_{\pm 0.14}$ | $\mathbf{52.68}_{\pm 3.20}$ | $87.36_{\pm 0.57}$ | $23.61_{\pm 3.08}$ |
| | BL-IST | DDN | $\mathbf{97.86}_{\pm 0.06}$ | $42.42_{\pm 2.66}$ | $87.81_{\pm 0.10}$ | $25.72_{\pm 2.15}$ |
| | BL-IST | SP | $97.70_{\pm 0.12}$ | $39.02_{\pm 2.36}$ | $\mathbf{87.89}_{\pm 0.19}$ | $\mathbf{25.84}_{\pm 1.79}$ |

## 6 CONCLUSION

In this work, we present a new method for enhancing the performance of trained image classifier networks. Our approach is particularly useful for small networks with relatively modest performance (i.e., 70-80%) typically deployed on edge devices. The method has two stages - first the training set samples for which the network gives incorrect answers are modified via corrective adversarial attacks so that the network now gives the correct answers. In the second stage, the network is refined via domain adaptation, using Deep CORAL, from the modified dataset to the original dataset. Experiments show substantial enhancements in performance on CIFAR datasets of over 5%, and 1% on CINIC-10.

Our experiments show that the adversarial correction approach is effective for refining quantized networks. Also, we observe that the adversarial correction enhances robustness to adversarial attack.

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
