## A    EXPERIMENTS SETUP (EXTENSION)

The efficiency of our method is evaluated on the basis of the reconstructed baseline result. The data transformation strategy we use in the data loading process should match the pre-trained model to produce the same result. In our experiments, we find that including random crops in data transformation can cause problems when applying adversarial attacks. Since random crop cuts a random region of the image, it causes the image we want to attack to differ from the original image. This can cause unsuccessful attacks and unstable performance of our method. It is worth mentioning that we need to shuffle the dataset $T'$ before conducting domain adaptation training. This ensures a mixture of training samples, including those from the original dataset for which the trained network gets the correct answers and the incorrect samples successfully perturbed.

In detail, we shuffle datasets before applying random crops and save the coordinates of the random crop regions in a file with the order of images in the training dataset if a random crop is included in the transformation process. For the correct dataset (a subset of the train dataset with correct predictions), we reload the random crop file and apply the recorded random crops in the domain adaptation training step. The correct dataset indices are saved following the order of the train dataset when we generate the model baseline so we can use these indexes to obtain random crop regions for the correct dataset. For the incorrect dataset (a subset of the train dataset predicted as incorrect) that needs to be attacked, we use the same strategy to obtain random crop indexes and apply them to images before applying attacks. Then we check the correction rate by comparing the labels before and after the attack to ensure that the labels are different. We also test the prediction of our baseline model on the correct dataset and altered dataset (incorrect dataset processed with attack). The accuracy for the correct dataset should be 100% and the accuracy for the altered dataset should be near 100% while using the DDN attack.

In the domain adaptation training step, we load the file only based on the current batch size. To be more precise, in the domain adaptation training, for each step, when the index is in the correct index list, the source image, target image, and source label are just the data from the train dataset. When the index is in the incorrect index list, the source image and source label processed with attacks should be taken from the altered dataset, and the target images are taken from the train dataset.

Memory overflow is another problem we face in the domain adaptation step. The loading of the entire file that stores all the attacked images can take a significant amount of memory, especially when the dataset is extremely large, such as ImageNet-1K. Therefore, we individually save these images into a *.npz* file, containing one attacked image and its correct label. Then, we design a customized dataloader that loads the correct images and the altered images as our **source** dataloader.

## B    CONSUMPTION OF COMPUTING RESOURCES

The adversarial correction process does take some time, but it's not as long as the training time. The computer resources needed to reproduce our experiment results are summarized in the Tab. 5. For large datasets, if we assume the performance decreases on larger datasets, we will have more corrections to do. However, the increase in time is expected to be near linear or slightly more than linear, not quadratic.

## C    PERFORMANCE OF BASELINE WITH FURTHER FINE-TUNING

We investigate the baseline performance with extended fine-tuning. Figure 3 shows our baseline training, validation, and test accuracy over 300 epochs for ResNet-34 on the CIFAR-100 dataset. We find that the performance plateaued at around epoch 60, and fine-tuning beyond 100 epochs did not yield further performance improvements.

Table 5: Compute resources utilized for each dataset with a batch size of 16. Note that the resources and time may vary slightly depending on the selected model.

| Dataset | Experiment | CPU | | | GPU | | | Computing Time |
|---|---|---|---|---|---|---|---|---|
| | | Number | Memory (GB) | Cores | Number | Memory (GB) | Type | |
| CIFAR-10 | Train Baseline | 1 | 3 | 32 | 1 | 2 | V100 | 2.0 min/epoch |
| | Adversarial Correction | 1 | 3 | 32 | 1 | 2 | V100 | 0.4 min/batch |
| | Deep CORAL (None attack) | 1 | 15 | 32 | 1 | 2 | V100 | 1.2 min/epoch |
| | Deep CORAL (Attack) | 1 | 20 | 32 | 1 | 2 | V100 | 1.2 min/epoch |
| CIFAR-100 | Train Baseline | 1 | 3 | 32 | 1 | 2 | V100 | 2.5 min/epoch |
| | Adversarial Correction | 1 | 3 | 32 | 1 | 2 | V100 | 0.5 min/batch |
| | Deep CORAL (w/o attack) | 1 | 15 | 32 | 1 | 2 | V100 | 1.5 min/epoch |
| | Deep CORAL (w/ attack) | 1 | 20 | 32 | 1 | 2 | V100 | 1.5 min/epoch |
| CINIC-10 | Train Baseline | 1 | 5 | 32 | 1 | 2 | V100 | 12 min/epoch |
| | Adversarial Correction | 1 | 30 | 32 | 1 | 2 | V100 | 0.5 min/batch |
| | Deep CORAL (w/o attack) | 1 | 38 | 32 | 1 | 2 | V100 | 6 min/epoch |
| | Deep CORAL (w/ attack) | 1 | 40 | 32 | 1 | 2 | V100 | 15 min/epoch |
| ImageNet-1K | Adversarial Correction | 1 | 40 | 32 | 1 | 12 | V100 | 1 min/batch |
| | Deep CORAL (w/o attack) | 1 | 45 | 32 | 1 | 12 | V100 | 300 min/epoch |
| | Deep CORAL (w/ attack) | 1 | 50 | 32 | 1 | 12 | V100 | 300 min/epoch |

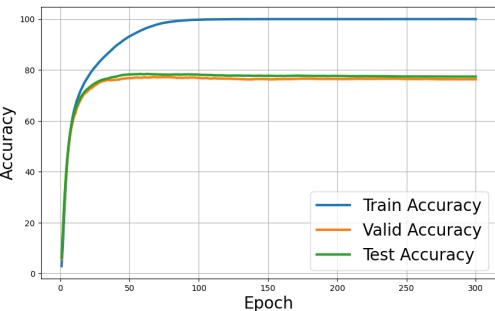

Figure 3: This graph shows our baseline training, validation, and test accuracy over 300 epochs for the ResNet-34 model on the CIFAR-100 dataset. The accuracy results are averaged over three random seeds.

# D    PERFORMANCE OF ADVERSARIAL CORRECTION (EXTENSION)

## D.1    ADVERSARIAL CORRECTION OF RESNETS

The comprehensive performance results of our pipeline on FP32 models are presented in Tab. 6, an extended version of Tab. 1. Additionally, the complete performance results of our pipeline on quantized Int8 models can be found in Tab. 7, which is an extended version of Tab. 3. The attacks all seem to have similar performance improvements, except for some directed attacks where we use different directions to reduce the effectiveness of incorrect classifications versus correct ones. Specifically, we can move away along the gradient of the highest probability class of incorrect samples to weaken the accuracy of the incorrect input (i.e., BIH attack) or move toward the true labels (i.e., VBI attack). Moving toward the true labels yields higher attack correction rates but similar performance compared to moving away along the gradient of the highest probability class. Table 6 also shows that increasing the number of attack iterations to increase the attack success rate for the VBI attack does not correlate closely with the correction rate. However, gradient-based attacks, such as DDN and VBI, have better overall performance than non-gradient-based attacks, such as salt and pepper.

The adversarial correction does indeed reduce the training loss. In Fig. 4b and Fig. 4d we can see that both targeted (VBI) and untargeted (LL) adversarial attacks can successfully reduce the logit level of the initially maximum probability incorrect label as compared with the logit level of the true label, resulting in correction.

Table 6: Accuracy (%) of FP32 baseline models (BL), which is fine-tuned on the CIFAR train domains, and accuracy of baselines after applying our approach (denoted as BL-IST) by using different attacks to generate adversarial domains. The data is reported as an average of three seeds.

| Model | Approach | Attack | CIFAR-10 | | | | | | CIFAR-100 | | | | | |
|---|---|---|---|---|---|---|---|---|---|---|---|---|---|---|
| | | | Corr. rate | $T'$ | Train | Valid | Test | Δ Acc | Corr. rate | $T'$ | Train | Valid | Test | Δ Acc |
| ResNet-18 (11.19M) | BL | - | - | - | 99.61±0.56 | 93.73±0.43 | 93.29±0.37 | - | - | - | 99.00±1.19 | 76.84±0.12 | 77.04±0.08 | - |
| | BL-IST | None | - | 99.96±0.04 | 99.69±0.36 | 95.91±0.20 | 95.57±0.13 | +2.28 | - | 98.86±0.94 | 98.84±0.98 | 80.14±0.47 | 80.27±0.74 | +3.23 |
| | BL-IST | LL | 55/176 | 100.00 | 99.80±0.28 | 96.17±0.08 | 95.93±0.15 | +2.64 | 70/451 | 100.00 | 99.20±0.92 | 80.99±0.21 | 80.93±0.46 | +3.90 |
| | BL-IST | BIH | 99/176 | 100.00 | 99.86±0.19 | 96.16±0.28 | 95.87±0.24 | +2.58 | 51/451 | 100.00 | 99.36±0.72 | 80.75±0.54 | **80.99±0.45** | **+3.96** |
| | BL-IST | VBI$_{iter1}$ | 121/176 | 100.00 | 99.59±0.46 | 96.09±0.22 | **95.97±0.12** | **+2.68** | 226/451 | 100.00 | 99.47±0.59 | 80.81±0.18 | 80.92±0.56 | +3.89 |
| | BL-IST | VBI | 175/176 | 100.00 | 99.97±0.04 | 96.19±0.15 | 95.77±0.06 | +2.48 | 446/451 | 100.00 | 99.80±0.20 | 80.37±0.98 | 80.54±0.80 | +3.50 |
| | BL-IST | DDN | 176/176 | 100.00 | 100.00 | 96.21±0.28 | 95.84±0.07 | +2.55 | 451/451 | 99.98±0.01 | 99.98±0.01 | 80.79±0.45 | 80.82±0.35 | +3.79 |
| | BL-IST | SP | 45/176 | 100.00 | 99.79±0.29 | 96.17±0.12 | 95.80±0.08 | +2.51 | 43/451 | 100.00 | 99.16±1.00 | 80.63±0.54 | 80.89±0.61 | +3.86 |
| ResNet-34 (21.30M) | BL | - | - | - | 99.43±0.67 | 94.71±0.05 | 94.22±0.06 | - | - | - | 94.36±2.24 | 78.12±0.79 | 78.41±0.10 | - |
| | BL-IST | None | - | 99.92±0.03 | 99.81±0.10 | 96.78±0.08 | 96.40±0.05 | +2.18 | - | 95.38±1.56 | 95.26±1.64 | 82.99±0.48 | 82.98±0.07 | +4.57 |
| | BL-IST | LL | 25/80 | 99.98±0.02 | 99.89±0.07 | 96.53±0.16 | 96.31±0.12 | +2.09 | 370/2538 | 100.00 | 96.05±1.39 | 83.13±0.08 | 82.69±0.12 | +4.28 |
| | BL-IST | BIH | 46/80 | 99.99±0.01 | 99.94±0.06 | 96.53±0.23 | 96.36±0.07 | +2.14 | 655/2538 | 100.00 | 97.31±1.19 | 83.04±1.19 | 83.31±0.06 | +4.90 |
| | BL-IST | VBI$_{iter1}$ | 53/80 | 99.99 | 99.97±0.01 | 96.62±0.07 | 96.26±0.12 | +2.04 | 1207/2538 | 100.00 | 97.40±0.92 | 83.39±0.44 | 83.11±0.23 | +4.70 |
| | BL-IST | VBI | 80/80 | 100.00 | 100.00 | 96.71±0.22 | 96.26±0.11 | +2.04 | 2490/2538 | 100.00 | 99.21±0.12 | 83.34±0.36 | 83.26±0.45 | +4.85 |
| | BL-IST | DDN | 80/80 | 100.00 | 100.00 | 96.71±0.22 | **96.71±0.05** | **+2.49** | 2538/2538 | 99.98±0.01 | 99.98±0.01 | 83.55±0.53 | **83.64±0.06** | **+5.23** |
| | BL-IST | SP | 23/80 | 99.98±0.01 | 99.90±0.09 | 96.52±0.12 | 96.22±0.05 | +2.00 | 118/2538 | 100.00 | 95.74±1.48 | 83.33±0.37 | 83.25±0.29 | +4.84 |
| ResNet-50 (23.57M) | BL | - | - | - | 99.81±0.14 | 95.36±0.36 | 94.32±0.59 | - | - | - | 98.81±0.13 | 80.01±0.65 | 79.74±0.19 | - |
| | BL-IST | None | - | 99.92±0.03 | 99.78±0.04 | 96.65±0.16 | **96.61±0.12** | **+2.29** | - | 99.84±0.01 | 98.34±0.38 | 83.70±0.14 | **83.89±0.22** | **+4.15** |
| | BL-IST | LL | 46/131 | 99.96±0.01 | 99.84±0.01 | 96.57±0.19 | 96.31±0.11 | +1.99 | 60/775 | 99.99±0.01 | 98.58±0.36 | 83.29±0.43 | 83.11±0.48 | +3.37 |
| | BL-IST | BIH | 69/141 | 99.95±0.03 | 99.85±0.02 | 96.41±0.11 | 96.11±0.16 | +1.79 | 261/775 | 99.98±0.02 | 98.69±0.26 | 82.86±0.50 | 83.03±0.43 | +3.29 |
| | BL-IST | VBI$_{iter1}$ | 79/231 | 99.99±0.01 | 99.89±0.02 | 96.61±0.10 | 96.18±0.26 | +1.86 | 304/775 | 99.99 | 97.55±0.50 | 80.61±0.16 | 83.00±0.17 | +3.26 |
| | BL-IST | VBI | 130/131 | 99.99±0.01 | 99.96±0.01 | 96.56±0.12 | 96.50±0.18 | +2.18 | 741/775 | 99.98±0.02 | 99.57±0.14 | 82.96±0.23 | 82.87±0.07 | +3.13 |
| | BL-IST | DDN | 131/131 | 99.97±0.04 | 99.97±0.04 | 96.61±0.27 | 96.35±0.12 | +2.03 | 775/775 | 99.98±0.01 | 99.98±0.01 | 83.29±0.42 | 83.03±0.07 | +3.29 |
| | BL-IST | SP | 17/131 | 99.97±0.01 | 99.82±0.01 | 96.61±0.22 | 96.30±0.15 | +1.98 | 45/775 | 99.99±0.01 | 98.58±0.35 | 83.00±0.19 | 83.25±0.32 | +3.51 |
| EfficientNetV2-M (52.99M) | BL | - | - | - | 99.96±0.06 | 97.66±0.13 | 97.15±0.14 | - | - | - | 99.88±0.08 | 86.63±0.73 | 86.88±0.46 | - |
| | BL-IST | None | - | 99.97±0.01 | 99.95±0.01 | 98.21±0.08 | 97.76±0.14 | +0.61 | - | 99.72±0.11 | 99.62±0.14 | 87.73±0.59 | 87.36±0.57 | +0.48 |
| | BL-IST | LL | 3/9 | 100.00 | 99.98 | 98.14±0.09 | 97.82±0.08 | +0.67 | 17/54 | 100.00 | 99.91±0.09 | 88.05±0.22 | 87.52±0.45 | +0.64 |
| | BL-IST | BIH | 6/9 | 100.00 | 99.99±0.01 | 98.20±0.09 | 97.82±0.09 | +0.68 | 23/54 | 99.97±0.05 | 99.91±0.09 | 87.96±0.32 | **88.00±0.10** | **+1.12** |
| | BL-IST | VBI$_{iter1}$ | 7/9 | 99.99±0.001 | 99.99±0.001 | 98.18±0.11 | 97.82±0.12 | +0.67 | 29/54 | 99.94 | 99.87±0.05 | 88.00±0.03 | 87.77±0.06 | +0.89 |
| | BL-IST | VBI | 8/9 | 99.99±0.001 | 99.99±0.01 | 98.13±0.12 | 97.80±0.04 | +0.65 | 46/54 | 99.95±0.01 | 99.88±0.004 | 88.09±0.18 | 87.76±0.16 | +0.88 |
| | BL-IST | DDN | 9/9 | 100.00 | 100.00 | 98.18±0.09 | **97.86±0.06** | **+0.71** | 54/54 | 99.92±0.04 | 99.92±0.04 | 87.98±0.18 | 87.81±0.10 | +0.93 |
| | BL-IST | SP | 4/9 | 99.99 | 99.98 | 98.13±0.05 | 97.70±0.12 | +0.55 | 18/54 | 99.95±0.03 | 99.87±0.07 | 87.85±0.04 | 87.89±0.19 | +1.01 |

Table 7: Accuracy (%) of quantized (Int8) ResNets of various sizes obtained after applying PTSQ on its baseline, and the accuracy of Int8 ResNets using our approach.

| Model | Approach | Attack | CIFAR-10 | | | | | | CIFAR-100 | | | | | |
|---|---|---|---|---|---|---|---|---|---|---|---|---|---|---|
| | | | Corr. rate | $T'$ | Train | Valid | Test | Δ Acc | Corr. rate | $T'$ | Train | Valid | Test | Δ Acc |
| ResNet-18 | BL | - | - | - | 99.61±0.56 | 93.73±0.43 | 93.29±0.37 | - | - | - | 99.00±1.19 | 76.84±0.12 | 77.04±0.08 | - |
| | PTSQ | - | - | - | 98.08±0.71 | 93.01±0.56 | 92.42±0.17 | - | - | - | 96.74±3.02 | 75.45±1.29 | 76.06±0.94 | - |
| | PTSQ-IST (bef. qt) | None | - | 99.96±0.04 | 99.43±0.02 | 95.88±0.26 | 95.59±0.07 | - | - | 99.80±0.12 | 97.13±0.57 | 80.34±0.59 | 80.18±0.39 | - |
| | PTSQ-IST (aft. qt) | None | - | 99.93±0.01 | 99.23±0.05 | 95.30±0.39 | 95.18±0.09 | +2.76 | - | 99.52±0.14 | 96.67±0.56 | 79.15±0.53 | 79.15±0.26 | +3.09 |
| | PTSQ-IST (bef. qt) | BIH | 158/736 | 100.00 | 99.53 | 96.07±0.06 | 95.85±0.19 | - | 300/1966 | 100.00 | 99.02±0.50 | 80.61±0.16 | 80.82±0.30 | - |
| | PTSQ-IST (aft. qt) | BIH | - | 99.99±0.01 | 99.46±0.04 | 95.53±0.29 | **95.48±0.18** | **+3.07** | - | 99.98±0.02 | 97.36±0.55 | 79.25±0.30 | 79.53±0.58 | +3.47 |
| | PTSQ-IST (bef. qt) | SP | 128/736 | 100.00 | 99.54±0.03 | 96.17±0.13 | 95.72±0.22 | - | 189/1966 | 100.00 | 97.17±0.62 | 80.40±0.45 | 81.07±0.23 | - |
| | PTSQ-IST (aft. qt) | SP | - | 100.00 | 99.48±0.02 | 95.46±0.20 | 95.29±0.06 | +2.93 | - | 99.98±0.01 | 97.31±0.66 | 79.27±0.73 | **79.79±0.49** | **+3.73** |
| ResNet-34 | BL | - | - | - | 99.43±0.67 | 94.71±0.05 | 94.36±0.26 | - | - | - | 94.36±2.24 | 78.12±0.79 | 78.41±0.10 | - |
| | PTSQ | - | - | - | 98.16±0.35 | 93.63±0.10 | 93.36±0.09 | - | - | - | 90.32±2.30 | 76.20±0.39 | 77.13±0.45 | - |
| | PTSQ-IST (bef. qt) | None | - | 99.97±0.02 | 99.44±0.15 | 96.57±0.15 | 96.28±0.13 | - | - | 99.09±0.10 | 93.08±0.23 | 82.94±0.29 | - | |
| | PTSQ-IST (aft. qt) | None | - | 99.92±0.04 | 99.31±0.17 | 96.19±0.10 | **96.08±0.20** | **+2.72** | - | 99.44±0.55 | 92.59±0.33 | 81.89±0.63 | 81.94±0.25 | +4.81 |
| | PTSQ-IST (bef. qt) | BIH | 250/771 | 100.00 | 99.45±0.11 | 96.61±0.19 | 96.33±0.15 | - | 689/4607 | 99.96±0.04 | 93.97±0.31 | 83.20±0.24 | 82.99±0.15 | - |
| | PTSQ-IST (aft. qt) | BIH | - | 99.97±0.02 | 99.39±0.12 | 96.29±0.08 | 96.05±0.07 | +2.69 | - | 99.99±0.01 | 93.77±0.33 | 82.19±0.14 | **82.18±0.20** | **+5.05** |
| | PTSQ-IST (bef. qt) | SP | 272/771 | 99.96±0.04 | 99.51±0.04 | 96.58±0.14 | 96.12±0.23 | - | 479/4607 | 100.00 | 93.80±0.52 | 83.13±0.54 | 82.90±0.19 | - |
| | PTSQ-IST (aft. qt) | SP | - | 99.95±0.05 | 99.45±0.05 | 96.25±0.06 | 95.83±0.19 | +2.47 | - | 100.00 | 93.55±0.53 | 81.99±0.24 | 82.12±0.20 | +4.99 |

(a) Incorr. spl, LL    (b) Corrected spl, LL    (c) Incorr. spl, VBI    (d) Corrected spl, VBI

Figure 4: The incorrect class (max) and true class logits change for uncorrected (a,c) and corrected (b,d) samples (spl) of CIFAR-100 after applying the corrective LL (a,b) and VBI (c,d) attacks on the ResNet-34. The vertical dashed lines indicate the mean values of incorrect class (max logit) and true class logits change.

## D.2 ADVERSARIAL CORRECTION OF TINY VISION TRANSFORMER

*Tiny vision transformer (TinyViT)* Wu et al. (2022) *baseline models on ImageNet-1K* Krizhevsky et al. (2012). We also selected TinyViT to evaluate our pipeline performance. We follow the same experiment settings as stated in Wu et al. (2022) to generate the TinyVit-21M baseline. TinyVit-21M is pre-trained on ImageNet-22K with the fast distillation framework using CLIP-ViT-L/14 Radford et al. (2021); Dosovitskiy et al. (2021) as the teacher, then finetuned on ImageNet-1K which has a total of 1,281,167 labeled train images, a validation set containing 50,000 images, and 1,000 object classes. We report validation accuracy instead of test accuracy, as the test dataset is unlabeled and reserved for Challenges.

We reproduce the TinyViT-21M Wu et al. (2022) using the same experiment setting, achieving 84.55% accuracy on the ImageNet-1K Krizhevsky et al. (2012) validation set and 83.2% accuracy when training with only 1,183,431 correctly classified samples. In the DDN attack case, we achieved 82.86% validation accuracy with only one epoch of training, which is 0.43% higher than the non-attack case.

### D.3 ADVERSARIAL CORRECTION WITH LONGER DOMAIN ADAPTATION PROCESS

Figure 5 demonstrates the performance of our pipeline on the training, validation, and test datasets of CIFAR-100 using ResNet-34. By applying Deep CORAL, we add an extra loss term (i.e., CORAL loss), which helps reduce overfitting by acting as a regularizing term. This approach is less likely to overfit compared to training the baseline model.

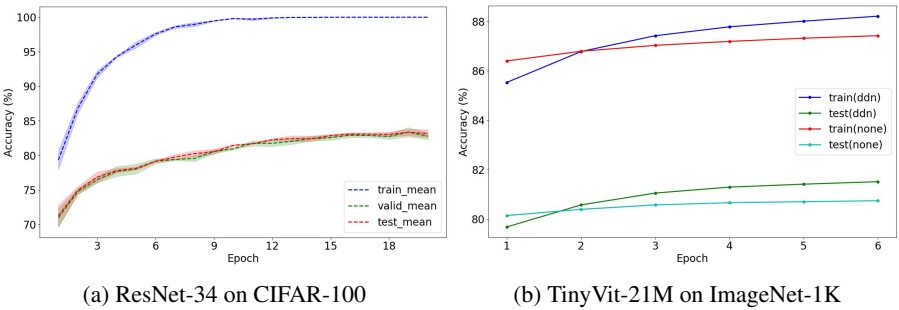

(a) ResNet-34 on CIFAR-100      (b) TinyVit-21M on ImageNet-1K

Figure 5: This graph shows our pipeline training, validation, and test accuracy with the DDN attack using (a) the ResNet-34 model on the CIFAR-100 dataset and (b) the TinyVit-21M model on the ImageNet-1K. The accuracy results are averaged over three random seeds.

### D.4 GRAD-CAM VISUALIZATION OF ADVERSARIAL CORRECTION

To help visualize the impact of the adversarial correction technique on misclassified images, we employ Gradient-weighted Class Activation Mapping (Grad-CAM) Selvaraju et al. (2017) to provide visual explanations. Grad-CAM utilizes gradient-based localization to identify important regions in an image that contribute to the prediction of the model concept. In our study, Fig. 6a is an example initially misclassified as "automobile" by ResNet-34. However, applying the DDN attack, the image can be correctly identified as a "horse". To better understand the differences between the Grad-CAM of the original (Fig. 6c) and its corrected image (Fig. 6d), we present a visualization in Fig. 6b. This visualization clearly illustrates that incorrect detection was primarily influenced by the contextual information surrounding the object rather than the object itself, demonstrating that by modifying the contextual information surrounding the image using adversarial attack, correct classification becomes possible.

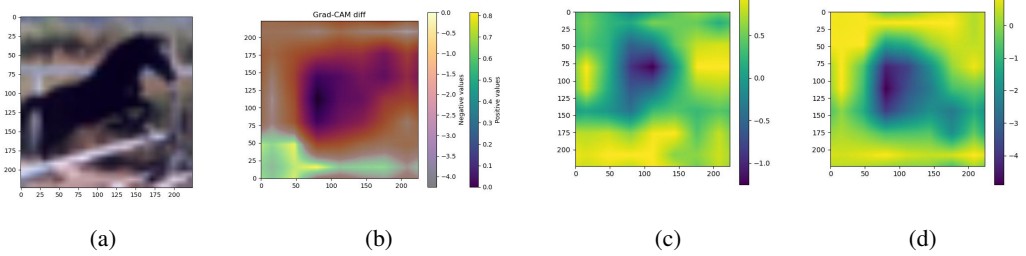

(a)      (b)      (c)      (d)

Figure 6: Evaluation of ResNet-34 on CIFAR-10 dataset. (a) misclassified images, (b) the difference between the Grad-CAM images for the original and adversarially corrected inputs using DDN attack. This illustrates the shift in focus of the network for the two images, (c) the Grad-CAM image for the original incorrect image, (d) the Grad-CAM image for the adversarially corrected image.

### D.5 VISUALIZATION OF ADVERSARIAL ATTACKS

In recent years, there has been an increasing amount of research aimed at developing techniques to deceive neural networks. These techniques, known as adversarial attacks, involve making malicious yet subtle changes in the input to fool the network Goodfellow et al. (2015a); Madry et al. (2018); Rony et al. (2019); Carlini & Wagner (2017); Moosavi-Dezfooli et al. (2016). Adversarial attacks make malicious yet subtle changes in the input to fool the network, as shown in Fig. 8. These changes are often imperceptible to the human eye, making it difficult to distinguish between the original image and the adversarially altered one. Our adversarial correction approach, on the other hand, takes training set images which the network classifies incorrectly, and alters (attacks) these images so that the network gives the right answer. Similarly, the differences between the misclassified image and the corrected image are not visually noticeable (see Fig. 10).

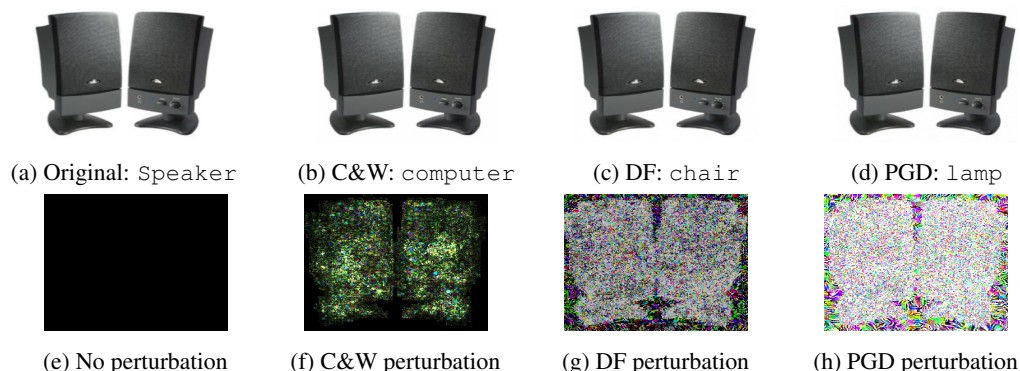

(a) Original: Speaker   (b) C&W: computer   (c) DF: chair   (d) PGD: lamp

(e) No perturbation   (f) C&W perturbation   (g) DF perturbation   (h) PGD perturbation

Figure 8: Subplots (a)-(d) show misclassified images of a speaker from the Amazon domain by ResNet-50 under C&W, DF, and PGD adversarial attacks. Subplots (e)-(h) show the corresponding perturbations generated under attacks, magnified by a factor of 500.

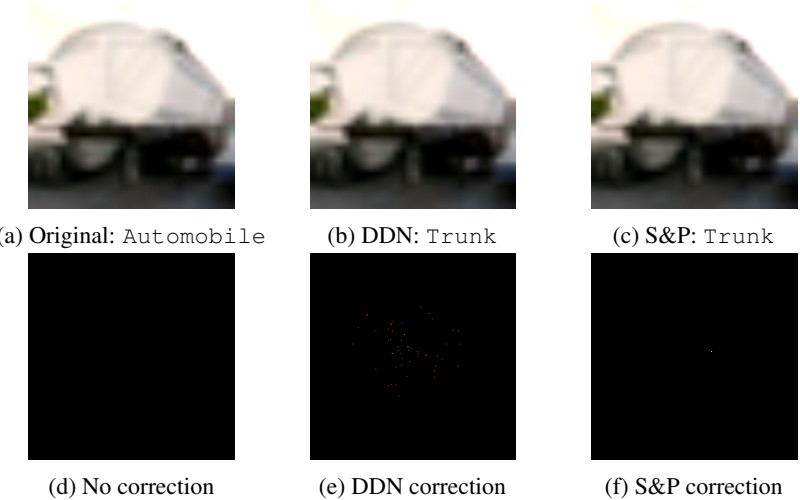

(a) Original: Automobile   (b) DDN: Trunk   (c) S&P: Trunk

(d) No correction   (e) DDN correction   (f) S&P correction

Figure 10: Subplots (a)-(c) show misclassified images of a trunk from the CIFAR-10 dataset by ResNet-34 and its corrected images under DDN and Salt and Pepper noise attacks. Subplots (d)-(f) show the corresponding perturbations generated under adversarial correction, magnified by a factor of 1000.

## D.6 ADVERSARIAL CORRECTION VS. ADVERSARIAL PERTURBATION

In the Feng and Tu theory Feng & Tu (2022), all that is needed in the first step of the IST is to perturb the input so as to reduce the loss. It is not necessary to actually change the input so as to have the network give the correct answer; all that is required is that the loss be reduced.

In the experiments shown in Tab. 1, we defined $T_a$ as the set of successfully corrected samples in step 3 of our adversarial correction approach. If we now consider $T_a$ to include all perturbed samples, whether the outputs are corrected or not, $T'$ will have the same size as the original training dataset. We refer to the network adapted using this variation as *BL-IST-A*. In our original approach, the accuracy of the original network on $T'$ reaches 100% when we consider only the successfully perturbed samples and the original correctly detected samples. Inspired by Shen et al. (2023), we can think of $T'$ as an *easy* dataset, given its 100% accuracy, while considering $T$ as a *hard* dataset. In Tab. 8, we observe a drop in performance improvement in BL-IST-A as compared to our first approach. This could be attributed to the adversarial perturbations increasing the loss rather than decreasing it, as compared with the baseline, for the uncorrected inputs. We conclude that we should only retain the corrected input samples.

Table 8: Accuracy (%) of ResNet FP32 baselines after applying our approach using the LL attack to generate adversarial domains for CIFAR datasets. Note that BL-IST-A is a refined approach in which $T_a$ in Step 3 incorporates all perturbed samples of $T_w$.

| Model | Approach | CIFAR-10 | | | CIFAR-100 | | |
|---|---|---|---|---|---|---|---|
| | | # $T'$ | Test | $\Delta$ Acc | # $T'$ | Test | $\Delta$ Acc |
| ResNet-18 | BL | - | 93.32 | - | - | 77.09 | - |
| | BL-IST | 44,972 | 95.77 | +2.45 | 44,879 | 80.48 | +3.39 |
| | BL-IST-A | 45,000 | 95.51 | +2.19 | 45,000 | 79.56 | +2.47 |
| ResNet-34 | BL | - | 94.24 | - | - | 78.53 | - |
| | BL-IST | 44,993 | 96.36 | +2.12 | 42,903 | 82.76 | +4.23 |
| | BL-IST-A | 45,000 | 96.18 | +1.94 | 45,000 | 80.81 | +2.28 |

## D.7 ADVERSARIAL CORRECTION VS. ADVERSARIAL TRAINING

While there are similarities in our approach and adversarial training, there are significant differences. Adversarial training takes training set images which the network classifies correctly and alters (attacks) these images so that the network gives the wrong answer. These images, with the correct label, are used to augment the training set. Our approach, on the other hand, takes training set images which the network classifies incorrectly, and alters (attacks) these images so that the network gives the right answer. Instead of augmenting the training set with these examples, we replace the initially wrong images with the adversarially corrected images. The advantage of our approach is that it improves accuracy as compared to standard adversarial training, since we are providing guidance for the network on how to do better on images it had trouble with. However, both approaches, as shown in the paper, provide robustness to adversarial attacks.

## D.8 ADVERSARIAL CORRECTION VS. CURRICULUM LEARNING

We compare our approach with three baselines: a vanilla baseline, and two curriculum learning (CL) baselines. In one CL baseline, we perform standard fine-tuning on easy data, then continue training on hard data. In the second CL baseline, we fine-tune on easy data and then train on the full training dataset.

Table 9 presents the performance of AdCorDA and the three baselines (fine-tuning and two CL approaches) on CIFAR datasets using ResNet-18. Our method, with any attack, outperforms all baselines, achieving 0.62% and 2.05% higher test accuracy than the better-performing CL baseline on CIFAR-10 and CIFAR-100, respectively.

It is worth noting that AdCorDA is more efficient and cost-effective in terms of both speed and the amount of labeled training data required. Specifically, the CL methods use double the labeled data compared to our approach, while Deep CORAL is an unsupervised method that does not rely on target domain labels during domain adaptation. Additionally, our domain adaptation process takes

only a few epochs, whereas the second stage of CL - continuing training - requires significantly more epochs.

Table 9: Compare accuracy (%) of ResNet-18 on CIFAR test datasets using of our AdCorDA approach with standard fine-tuning baseline and different types of curriculum learning.

| Approach | Attack | CIFAR-10 | CIFAR-100 |
|---|---|---|---|
| BL | - | 93.29 | 77.04 |
| CL (easy → hard) | - | 95.05 | 75.25 |
| CL (easy → train) | - | 95.31 | 78.94 |
| BL-IST | None | 95.57 | 80.27 |
| BL-IST | LL | 95.93 | 80.93 |
| BL-IST | BIH | 95.87 | 80.99 |
| BL-IST | VBI | 95.77 | 80.54 |
| BL-IST | DDN | 95.84 | 80.82 |
| BL-IST | SP | 95.80 | 80.89 |

## D.9 ADVERSARIAL CORRECTION AGAINT NOISY LABELED DATASETS

We investigate the performance of our AdCorDA approach under 20% symmetrical noisy labeled setting on CIFAR datasets. We first established new CIFAR baselines using standard fine-tuning, then applied our AdCorDA approach. Table 10 compares the performance of our approach using ResNet-34 on CIFAR datasets with and without noisy labels. The results demonstrate that our approach remains highly effective under noisy label settings, outperforming the noisy CIFAR-10 baseline by 4.77% in the none case and the noisy CIFAR-100 baseline by 8.77% in the DDN attack case.

Table 10: Accuracy (%) of baseline and our AdCorDA approach on CIFAR datasets using ResNet-34 with and without 20% noisy labels.

| Dataset | Approach | Attack | Corr. Rate | Train | Test |
|---|---|---|---|---|---|
| CIFAR-10-noisy | BL | - | - | 76.26 | 91.07 |
| | BL-IST | None | - | 78.98 | **95.84 (+4.77)** |
| | BL-IST | DDN | 10685/10685 | 78.22 | 93.40 (+2.33) |
| CIFAR-100-noisy | BL | - | - | 72.92 | 72.77 |
| | BL-IST | None | - | 75.07 | 81.31 (+8.54) |
| | BL-IST | DDN | 12188/12188 | 77.40 | **81.54 (+8.77)** |
| CIFAR-10 | BL | - | - | 99.43 | 94.22 |
| | BL-IST | None | - | 99.81 | 96.40 (+2.18) |
| | BL-IST | DDN | 80/80 | 100.00 | **96.71 (+2.49)** |
| CIFAR-100 | BL | - | - | 94.36 | 78.41 |
| | BL-IST | None | - | 95.26 | 82.98 (+4.57) |
| | BL-IST | DDN | 2538/2538 | 99.97 | **83.64 (+5.23)** |