# OpenReview forum: "AdCorDA: Classifier Refinement via Adversarial Correction and Domain Adaptation"
_ICLR.cc/2025/Conference — Submitted to ICLR 2025_

### Official Review · Reviewer_sAF4 · 2024-11-03

**Soundness:** 3
**Presentation:** 3
**Contribution:** 3
**Rating:** 6
**Confidence:** 4

**Summary:**

This paper proposed AdCorDA, a classifier refinement framework consisting of two stages. In the first stage, those incorrectly predicted examples are modified by adversarial attacks to move to correct predictions. In the second stage, the model is adapted from the unmodified examples and modified examples back to the original domain. Through extensive experiments, AdCorDA is proven to improve the performance of the normally trained baseline classifier. Besides, it can also improve the robustness under adversarial attacks.

**Strengths:**

- The paper is well-written and easy to follow.
- The idea of the two-stage correction-then-adaptation framework is intuitive and reasonable.
- The proposed method is effective and evaluated by extensive experiments.

**Weaknesses:**

- I noticed that you set the attack budget $\epsilon=5e-4$, which is much smaller than the commonly used $\epsilon=8/255$. It is not convincing to claim that "our method provides significant robustness to adversarial attacks" under such a weak attack where the baseline model can still get over 15% accuracy.
- The proposed method is only compared with a vanilla baseline. No other curriculum learning or domain adaptation methods are compared.

**Questions:**

- It seems that AdCorDA is only able to refine neural networks that don't get a very high training accuracy. Let's consider a highly overfitting scenario. Imagine that there is a highly over-parameterized neural network trained on a small-scale dataset, and this network is capable of overfitting the whole dataset, getting 100% training accuracy. This is likely to happen, as existing research like [1] already proved that over-parameterized neural networks can easily get 100% training accuracy even on randomly labeled CIFAR-10. In this case. no training examples are incorrectly predicted. How does AdCorDA work?
- In Tab. 4 How are the baseline models trained? Via normal training or adversarial training?
- Will stronger attacks help the first stage? I guess stronger attacks will increase the success rate of label correction.
- In Tab. 1, does BL-IST with Attack=None means just domain adaptation?

[1] Zhang, C., Bengio, S., Hardt, M., Recht, B., & Vinyals, O. (2021). Understanding deep learning (still) requires rethinking generalization. Communications of the ACM, 64(3), 107-115.

---

> ### Author Response · Authors · 2024-11-20
> **Response to comment**
>
> Thank you for your comment.
>
> 1) For weakness 1: We acknowledge that describing our method as providing significant robustness may be an overstatement. Our method provides some robustness as a by-product, but it is not intended to be state-of-the-art. This robustness is a side effect of our fine-tuning process.
>
>     Initially, we evaluated adversarial robustness under the regular setting with an attack strength of 8/255. However, both the baselines and our approach had very low accuracy under this setting. Therefore, we progressively reduced the attack strength. At $\epsilon$=5e-4, we observed a clear performance gap between our method and the baselines, making it an appropriate choice for analysis.
>
> 2) For weakness 2: We compare our approach with three baselines: a vanilla baseline, and two curriculum learning (CL) baselines. In one CL baseline, we perform standard fine-tuning on easy data, then continue training on hard data. In the second CL baseline, we fine-tune on easy data and then train on the full training dataset.
>
>     Table 9 in the Appendix D.8 of the supplementary material presents the performance of AdCorDA and the three baselines (fine-tuning and two CL approaches) on CIFAR datasets using ResNet-18. Our method, with any attack, outperforms all baselines, achieving 0.62\% and 2.05\% higher test accuracy than the better-performing CL baseline on CIFAR-10 and CIFAR-100, respectively.
>
>     It is worth noting that AdCorDA is more efficient and cost-effective in terms of both speed and the amount of labeled training data required. Specifically, the CL methods use double the labeled data compared to our approach, while Deep CORAL is an unsupervised method that does not rely on target domain labels during domain adaptation. Additionally, our domain adaptation process takes only a few epochs, whereas the second stage of CL - continuing training - requires significantly more epochs.
>
> 1) For question 1: We agree that AdCorDA is not designed to improve networks with very high training accuracy, such as highly over-parameterized networks that overfit small datasets (e.g., 100\% training accuracy on CIFAR-10). Our approach targets underparameterized networks, such as those used in edge devices, which typically do not achieve such high training accuracy. In overfitting scenarios, like the one you mentioned, AdCorDA would not be effective.
>
>     For overfitting, techniques like data augmentation (e.g., adding noise to images) are often used to increase dataset diversity, but they complicate the correction process, as each augmentation needs to be tracked. Our method is intended for scenarios where networks are smaller or less prone to overfitting, where it can offer improvements in robustness.
>
> 2) For question 2: We obtain the baseline models via normal training, i.e., standard fine-tuning from a pre-trained model.
>
> 3) For question 3: We aim to achieve a high correction rate to enhance the effectiveness of our approach, though this can vary depending on the attack method used. Yes, you are correct. Some methods require stronger or more aggressive attacks to reach such high levels of correction.
>
>     In our experiments, the DDN attack gives 100\% correction rate. Furthermore, as shown in Fig. 2(b) of our paper, as the number of corrected incorrect samples increases, we obtain nearly linear improvements in accuracy gain.
>
> 4) For question 4: Yes, as shown in Fig. 1 of the paper,  BL-IST represents the "none" case, which applies only the domain adaptation $T' \rightarrow T$ where $T'$ contains only $T_c$).

---

> ### Author Response · Authors · 2024-11-29
> **Follow-Up on Reviewer Comments**
>
> Thank you for taking the time to review our paper and provide valuable feedback. I kindly request your feedback on our responses to ensure we have the opportunity to further clarify any points or address additional concerns you may have. Your insights are greatly appreciated. Thank you again for your time and efforts.

---

### Official Review · Reviewer_eHY1 · 2024-11-03

**Soundness:** 2
**Presentation:** 3
**Contribution:** 2
**Rating:** 3
**Confidence:** 4

**Summary:**

In this paper, the authors propose a method to refine a pre-trained model by incorporating adversarial correction and domain adaptation. The method is motivated by prior work which shows that applying domain adaptation from easy to hard in curriculum learning can speed up training. So they create an easy dataset using adversarial correction and then align the synthetic dataset with the original dataset by aligning the second-order covariance matrices. They conducted some experiments to validate the superiority of their method.

**Strengths:**

* The authors introduce a simple yet effective way to refine a pre-trained classifier.
* They conduct fair experiments to validate the performance.

**Weaknesses:**

* I suggest that the authors show a more intuitive figure to visualize the framework that includes the images and labels in the original dataset and also the corrected images. This will help the readers to gain more intuition for your method.
* The authors combine two existing techniques to get the framework without innovation. The adversarial attack or correction method and the domain adaptation method used by the authors are proposed by prior work. And the adopted domain adaptation method here is a very old and simple method which is proposed eight years ago. Considering there were so many effective domain adaptation methods proposed in the recent few years, why don't you use other domain adaptation methods to further improve the performance?
* In Section 3.3, the authors align the features of the weak classifier on the original dataset and the synthetic dataset. Considering that the difference between the original and the synthetic datasets is the corrected part, can we omit the correctly classified samples and only minimize the covariance difference for the adversarially corrected sample and the misclassified sample?
* How do you choose the hyper-parameters such as $\lambda,\epsilon$? Does your method work robustly for other choices of hyper-parameters? If not, how do you choose them?

**Questions:**

* Which attack (correction) method do you use? In the introduction part, the authors say that they use the adversarial attack method targeting the true label to get an easier dataset where the classifier has 100% accuracy. However, the BI method generates more difficult samples for the classifier. Do you use this method? And what's the role of auto-attack in 3.2?

---

> ### Author Response · Authors · 2024-11-20
> **Response to weaknesses 1 and 2**
>
> Thank you for your comment.
> 1) For weakness 1: Including the original dataset and corrected images is not particularly useful, as the visual differences are imperceptible. Adversarial attacks make malicious yet subtle changes in the input to fool the network, as shown in Fig. 8 of Appendix D.5 of supplementary material. These changes are often imperceptible to the human eye, making it difficult to distinguish between the original image and the adversarially altered one. Our adversarial correction approach, on the other hand, takes training set images which the network classifies incorrectly, and alters (attacks) these images so that the network gives the right answer. Similarly, the differences between the misclassified image and the corrected image are not visually noticeable (see Fig. 10 of Appendix D.5).
>
>     In Appendix D.4 of the supplementary material - Grad-GAM visualization of adversarial correction, we employ Grad-CAM to provide visual explanations about the effectiveness and impact of the adversarial correction technique on misclassified images. This visualization clearly illustrates that the incorrect detection was primarily influenced by the surrounding contextual information rather than the object itself. This demonstrates that by modifying the surrounding contextual information of the image using the adversarial attack, correct classification becomes possible.
>
>     Furthermore, we demonstrate that adversarial correction effectively reduces the training loss, as shown in Fig. 4b and Fig. 4d in Appendix D.1 of the supplementary material. Both targeted (VBI) and untargeted (LL) adversarial attacks can successfully reduce the logit level of the initially maximum probability incorrect label as compared with the logit level of the true label, resulting in correction.
>
> 2) For weakness 2: We believe that our contributions are novel, and would like to point out other reviewers comment regarding novelty below:
>     - reviewer uPRK - ``The method for refining the classifier by adversarial correction and domain adaptation is novel to me".
>     - reviewer GYy1 - ``I really liked the idea of using adversarial example methods for fixing hard examples. I really liked the paper both in terms of novelty and contributions".
>
>     While adversarial attack and domain adaptation are established techniques, our innovation lies in how we combine them to improve performance of classifiers. Specifically, we are not proposing new adversarial attack methods or domain adaptation approaches but instead introducing a novel way to reformulate the fine-tuning problem as a domain adaptation problem. This involves identifying two domains with distinct distributions, defined through a curriculum-based approach. The domains consist of correctly classified and incorrectly classified samples. Our adversarial correction technique is designed to increase the number of samples in the first distribution - the correctly classified samples -— thereby enhancing the classifier's performance.
>
>     Additionally, we propose the concept of adversarial correction, which differs from the adversarial attacks typically used in adversarial training. Adversarial training takes training set images which the network classifies correctly and alters (attacks) these images so that the network gives the wrong answer. These images, with the correct label, are used to augment the training set. Our approach, on the other hand, takes training set images which the network classifies incorrectly, and alters (attacks) these images so that the network gives the right answer. Instead of augmenting the training set with these examples, we replace the initially wrong images with the adversarially corrected images. The advantage of our approach is that it improves accuracy as compared to standard adversarial training, since we are providing guidance for the network on how to do better on images it had trouble with.
>
>     We agree that exploring other domain adaptation methods could further showcase the effectiveness and generalization of our AdCorDA pipeline. As a starting point, we chose CORAL loss due to its simplicity, efficiency, and suitability for our current objectives. CORAL loss focuses on aligning second-order statistics between domains, which directly addresses the feature misalignment challenge central to our work.

---

> ### Author Response · Authors · 2024-11-20
> **Response to weaknesses 3-4 and question**
>
> 3) For weakness 3: If we only include samples from the incorrect or the corrected versions and remove all the common ones, the "none" case of our method would not exist, as the source domain (which contains only correct samples) would be absent. For the attack case, while the CORAL loss would remain the same, the cross-entropy loss would be affected significantly. This might improve performance on the incorrect samples but would likely degrade performance on the correct ones. Therefore, it's important to include some samples from the common parts of the distribution to achieve comparable performance. This also ensures there are enough training samples for the domain adaptation process.
>
>     One study points out that in domain adaptation tasks, especially when using CORAL loss, the alignment is more effective when the source and target domains have overlapping characteristics (Wang and Kang 2021, Rahman et al. 2019). This overlap facilitates better alignment of the feature distributions, leading to improved performance in transferring knowledge across domains. In cases where there's little overlap or large domain shifts, the alignment might struggle to capture the necessary nuances between the domains, reducing effectiveness.
>
>     Similarly, another study highlights the role of synthetic data in augmenting domain adaptation techniques like Deep CORAL, where the overlap between synthetic (source) and real (target) domains plays a crucial role in improving feature adaptation (Das et al. 2021). The use of synthetic data alongside domain adaptation methods also supports this view that better overlap leads to more successful domain adaptation, as it minimizes the distribution mismatch between the source and target.
>
>     Thus, for methods like CORAL loss to work optimally, ensuring that the source and target domains share some degree of distribution overlap is important for effective alignment and performance improvement.
>
>
>     Reference
>
>     [1] Z.-Y. Wang and D.-K. Kang, “P-norm attention deep coral: Extending correlation alignment using attention and the p-norm loss function,” Applied Sciences, vol. 11, no. 11, 2021.
>
>     [2] M. M. Rahman, C. Fookes, M. Baktashmotlagh, and S. Sridharan, “On minimum discrepancy estimation for deep domain adaptation,” CoRR, vol. abs/1901.00282, 2019.
>
>     [3] T. Das, R. Bruintjes, A. Lengyel, J. van Gemert, and S. Beery, “Domain adaptation for rare classes augmented with synthetic samples,” CoRR, vol. abs/2110.12216, 2021
>
> 4) For weakness 4: The weight of the CORAL loss $\lambda$ is a weight that trades off the adaptation with classification accuracy on the source domain (Sun \& Saenko (2016) in our paper). Following the experimental setup of the original Deep CORAL, $\lambda$ is a hyperparameter chosen such that, at the end of training, the classification loss and CORAL loss are approximately equal. This choice is reasonable, as it ensures the feature representation is both discriminative and minimizes the distance between the source and target domains.
>
>     As indicated in the Sec. 4 experimental setup of our paper, For the DDN and SP attacks, we use the default hyper-parameters (including $\epsilon$, step size, etc.) provided by the Foolbox framework (Rauber et al. (2017; 2020) in our paper). The BI and LL attacks are applied according to the experimental setting outlined in (Kurakin et al. (2017) in our paper) with one iteration step.
>
>     We set $\epsilon$=5e-4 for all AutoAttack experiments. Initially, we evaluated adversarial robustness under the regular setting with an attack strength of 8/255. However, both the baselines and our approach had very low accuracy under this setting. Therefore, we progressively reduced the attack strength. At $\epsilon$=5e-4, we observed a clear performance gap between our method and the baselines, making it an appropriate choice for analysis.
>
> 1) For question: The attack (correction) methods used in our experiments are described in detail in Sec. 3.2 of our paper. We are using the BI technique to shift the image to one which the network gives the correct answer. While standard adversarial attacks aim to degrade performance, as shown in the robustness experiments, adversarial correction takes the opposite approach. Instead of creating more challenging samples, adversarial correction applies adversarial attacks to subtly modify misclassified images so the network can correctly classify them. The focus is not on creating inherently more difficult or easier samples but on enabling accurate classification through targeted adjustments. The role of AutoAttack is to generate adversarial samples for testing adversarial robustness.

---

> ### Author Response · Authors · 2024-11-29
> **Follow-Up on Reviewer Comments**
>
> Thank you for taking the time to review our paper and provide valuable feedback. I kindly request your feedback on our responses to ensure we have the opportunity to further clarify any points or address additional concerns you may have. Your insights are greatly appreciated. Thank you again for your time and efforts.

---

### Official Review · Reviewer_GYy1 · 2024-11-04

**Soundness:** 4
**Presentation:** 4
**Contribution:** 4
**Rating:** 6
**Confidence:** 3

**Summary:**

Summary: This paper introduces AdCorDA, a method for refining pretrained classifiers, particularly targeting smaller networks on edge devices. The proposed technique consists of two stages: (1) adversarial correction, where adversarial attacks adjust misclassified samples to be correctly labeled by the network, creating a synthetic training dataset, and (2) domain adaptation, where the network is adapted from this synthetic dataset back to the original dataset to bridge any distribution shift. Experiments on CIFAR-10, CIFAR-100, and CINIC-10 demonstrate that AdCorDA provides performance improvements, with gains of 5% on CIFAR-100 and 1% on CINIC-10, while also enhancing the robustness of both full-precision and quantized networks against adversarial attacks.

**Strengths:**

I really liked the idea of using adversarial example methods for fixing hard examples. I really liked the paper both in terms of novelty and contributions.

The paper was well written, and easy to follow.

**Weaknesses:**

1) I believe that the paper could really benefit from further experimentation. Even though CINIC 10 is kind of a bridge between ImageNet and CIFAR datasets, I still do not see the limiting factor from scaling up to imagenet experiments. I also wonder if the effectiveness  of  this method limited to CNNs. I would like to ask authors to try ImaeNet on Transformer architecture and report the results or simply explain why such experiments would not be practical/relevant.

**Questions:**

1) As I mentioned previously, I would really love to ask for extending the experiments to larger datasets like ImageNet and their structure to transformer based models.

2) There are several studies that show that datasets like CIFAR and IMAGENET come with a noticeable number of incorrectly labeled samples. Intuitively in a perfect classifier, such noisy labeled samples should be predicted to have the correct label, which is different from the incorrect class specified in the labels from the dataset. I wonder how much of the robustness is gained from the removal of such images? Or maybe, the authors could visualize all the examples that start in Tw but do not end up in Ta. Do most of such examples have the wrong label assigned to them in the dataset? How much of such images end up in Ta?

3) I cannot help but wonder regarding the effectiveness of your method and the noisy label settings. I was wondering if the authors have considered trying their method against noisy label settings for a basic setting like a resnet34 trained on 20% Symmetrical Noisy Labeled setting on CIFAR-10/100.

If my questions/concerns are addressed, I will be happy to increase my score.

---

> ### Author Response · Authors · 2024-11-20
> **Response to comment**
>
> Thank you for your comment.
> 1) For weakness: Our approach focuses on small networks to optimize performance on edge devices.
> 2) For question 2: We need to speculate on this. We expect that the percentage of noisy labels is low and uniformly distributed across all classes. Identifying noisy-labeled samples, even for datasets like CIFAR-100, is a challenging and time-consuming task that we haven't undertaken. For ImageNet, this would be even more difficult, as it would likely require human labeling and still be prone to errors. This would involve domain experts, which can be challenging to access. Determining noisy samples is inherently complex. This is an interesting point that we hadn't considered before, but we're starting to explore it now.
> 3) For question 3: Consider the samples that are classified as incorrect due to label noise - where the labels are incorrect, but the data itself is correct. We think these samples are more easily perturbed because the added label noise makes them inherently unstable. For samples without label noise that are still classified as incorrect, our approach would perform as it normally does. The interesting part is how it works on the ones that we're correcting that had the label noise. So we're now moving them into the incorrect category, whereas they were originally correct. This shift can lead to lower training and test accuracy, as the model learns from incorrect labels.
>
>     As you suggested, we applied a 20\% symmetrical noisy label setting to both the CIFAR-10 and CIFAR-100 datasets. We first established new CIFAR baselines using standard fine-tuning, then applied our AdCorDA approach. Table 10 in the Appendix D.9 in the supplementary material compares the performance of our approach using ResNet-34 on CIFAR datasets with and without noisy labels. The results demonstrate that our approach remains highly effective under noisy label settings, outperforming the noisy CIFAR-10 baseline by 4.77\% in the none case and the noisy CIFAR-100 baseline by 8.77\% in the DDN attack case.

---

> ### Author Response · Authors · 2024-11-29
> **Follow-Up on Reviewer Comments**
>
> Thank you for taking the time to review our paper and provide valuable feedback. I kindly request your feedback on our responses to ensure we have the opportunity to further clarify any points or address additional concerns you may have. Your insights are greatly appreciated. Thank you again for your time and efforts.

---

### Official Review · Reviewer_uPRK · 2024-11-06

**Soundness:** 2
**Presentation:** 3
**Contribution:** 2
**Rating:** 3
**Confidence:** 4

**Summary:**

This paper proposes a method for refining pre-trained classifiers through adversarial correction and domain adaptation. Adversarial correction adds adversarial perturbations to misclassified samples to correct the network's predictions. Then an existing domain adaptation method Deep CORAL is applied to adapt from the corrected dataset back to the original data. Experiments on CIFAR-10, CIFAR-100, and CINIC-10 show consistent accuracy gains and improved adversarial robustness. The method is claimed to be friendly on edge devices.

**Strengths:**

1. The method for refining the classifier by adversarial correction and domain adaptation is novel to me.
2. Experiment results are pretty promising.

**Weaknesses:**

1. The insight behind the method is not clear. The authors did not make a strong effort to demonstrate why their method is effective. No theoretical proof is provided. And it is not clear what motivates this method. It is more like a heuristic without sufficient causal explanation.
2. It is questionable whether the adversarial correction is really necessary. From the experiments, the performance with no attack is very close to the methods with attacks, sometimes even slightly better. The authors did not demonstrate why they need adversarial correction. I am confusing the motivation for introducing this stage in your method.
3. Only compare with baselines without adversarial attack and domain adaptation. I wonder if it's necessary to compare with other related methods, like methods for edge devices? It is ambiguous about the advantages of the proposed method, although the authors show some accuracy improvements. If you claim to achieve better accuracy, you need to compare with more classification methods. If you claim to achieve better adversarial robustness, you need to compare with more adversarial learning methods. If you claim efficiency and resource-friendly on the edge devices, you need to compare methods implemented on edge devices.
4. In Table 5 of the supplementary material, the running time and memory are reported. The results are somehow misleading. The time of adversarial correction is reported by batch while the other steps are reported by epoch. Thus adversarial correction should introduce much more computing time. I am concerned whether this method can really be implemented on edge devices, given such an extremely large additional computing time.

**Questions:**

1. Why do you use pretrained models, like ImageNet pretrained, Place365 pretrained? Does training from scratch for your method and baselines produce different results?
2. What do the choices of adversarial attack hyperparameters like epsilon, iteration step, step size, etc, influence in your method?

---

> ### Author Response · Authors · 2024-11-20
> **Response to weaknesses**
>
> Thank you for your comment.
> 1) For weakness 1: We do not have theoretical proof, but we do have a clear motivation supporting this idea. We highlight our motivation in the first paragraph of Sec. 5 of our paper. Inspired by curriculum learning, we consider data in different difficulty levels as data with different distributions, i.e., in distinct domains. Therefore, instead of training from an easy to a hard curriculum, we can transfer knowledge from one domain of the dataset (e.g., source domain - an easy domain with 100\% accuracy) to a related but different domain (e.g., target domain - hard domain) within the dataset. This is inspired by the work in ((Shen et al. 2023) in our paper), who used domain adaptation in this way in a standard curriculum learning process. They found that this form of curriculum learning was much faster than standard curriculum learning. Our approach differs in two significant ways from the method presented in ((Shen et al. 2023) in our paper): first, it does not require an external scoring function to create the easy/hard curriculum, instead using the ground truth fidelity. We can transfer knowledge from one domain of the dataset (e.g., source domain - an easy domain with 100\% accuracy) to the original training dataset (e.g., target domain - hard domain) to which the model needs to adapt. Second, we enhance the source domain by adding the adversarial corrected data samples, thereby improving the domain adaptation.
>
>     The removal of the incorrect samples and the addition of the corrected samples provide a purer representation of the domain that the initial network does well on, thereby enhancing the effectiveness of the subsequent domain adaptation step. Indeed, even just removing the incorrect samples, without adding the adversarial corrections, provides a significant benefit to the domain adaptation step.
>
> 2) For weakness 2: The adversarial correction (i.e., attack case) helps improve model performance in most cases because it gives more distribution shift between the correct and incorrect samples compared to the none case. This is because there are more examples of correct samples (i.e., Ta in Fig. 1 of our paper) in the attack case. While the effect might be minor, with the source domain probably only about 5\% larger in the attack case, this increase yields improvement in many cases.
>
> 3) For weakness 3: Our approach can be applied to enhance other classification methods. While we don't claim our approach represents state-of-the-art adversarial robustness, it does provide some degree of robustness. This adversarial robustness is a by-product rather than the main focus of the paper.
>
>     You've mentioned that we need to compare methods that implemented on edge devices. Are you asking if this approach could be adapted for other architectures, such as MobileNet or EfficientNet? Could you clarify this question a bit? Thank you.
>
> 4) For weakness 4: The adversarial correction process is performed only once on the misclassified training samples from the baselines. There is no need to do the correction on every epoch of the domain adaptation stage. Therefore, the time for adversarial correction is reported per batch. For example, on the CINIC-10 dataset, ResNets have an average of 4,171 misclassified samples. With an adversarial correction time of 0.5 minutes per batch, the correction process takes approximately 30 minutes. However, performing standard fine-tuning on CINIC-10 requires around 1200 minutes.
>
>     The adversarial correction process does take some time, but it’s not as long as the training time. For large datasets, if we assume the performance decreases on larger datasets, we will have more corrections to do. However, the increase in time is expected to be near linear or slightly more than linear, not quadratic.

---

> ### Author Response · Authors · 2024-11-20
> **Response to questions**
>
> 1) For question 1: Our focus is on a fine-tuning process. We start with a pretrained model, and our domain adaptation also begins from this pretrained model. Training from scratch wouldn't yield any different result since the pretrained model was originally trained from scratch as well.
>
>     If you're suggesting we perform AdCorDA step-by-step from the very first epoch (train for one epoch, apply adversarial correction, then adapt, and continue training), that's not our approach. This would be very costly and time-consuming, and likely ineffective. In the early stages, the model hasn't been trained enough to accurately differentiate between correct and incorrect predictions and adversarial correction would be very time-consuming due to the low training accuracy in the early epochs resulting in many samples needing to be corrected.
>
> 2) For question 2: As indicated in the Sec. 4 experimental setup of our paper, for the DDN and SP attacks, we use the default hyper-parameters (including epsilon, step size, etc.) provided by the Foolbox framework (Rauber et al. (2017; 2020) in our paper). The BI and LL attacks are applied according to the experimental setting outlined in (Kurakin et al. (2017) in our paper) with one iteration step.

---

> ### Author Response · Authors · 2024-11-29
> **Follow-Up on Reviewer Comments**
>
> Thank you for taking the time to review our paper and provide valuable feedback. I kindly request your feedback on our responses to ensure we have the opportunity to further clarify any points or address additional concerns you may have. Your insights are greatly appreciated. Thank you again for your time and efforts.

---

### Author Response · Authors · 2024-12-02
**No interaction from reviewers on our responses**

Dear Area Chair,
There has been no interaction with the reviewers, despite us offering detailed responses, including an additional experiment they asked for, and clarifications of some technical points. Could you please encourage the reviewers to interact with us and offer a round of responses?
Regards,
The Authors

---

### Meta-Review · Area_Chair_9VSE · 2024-12-20

**Metareview:**

This paper introduces AdCorDA, a two-stage method for refining pretrained classifiers through adversarial correction and domain adaptation. The authors conducted experiments on standard and weight-quantized neural networks to demonstrate the effectiveness of AdCorDA. The main idea builds upon two existing techniques: replacing incorrectly predicted training examples with their adversarial counterparts and applying CORAL for domain adaptation. During the rebuttal phase, the authors provided further explanations to address the reviewers' concerns. The Area Chair engaged in additional discussions with reviewers regarding the concerns of the existing submission. However, several concerns remain. First, the existing experiments do not sufficiently support the claims made in the paper, such as the lack of experiments on diverse datasets to demonstrate improvements in accuracy, which is the primary focus of this work. Additionally, some reviewers pointed out that the improvement gains are relatively limited on certain evaluated models, such as EfficientNet. Furthermore, a more detailed ablation study of each component of the algorithm, such as adversarial correction and the hyperparameters, would help the audience better understand how the algorithm works. Lastly, since the work combines adversarial correction and domain adaptation in a general way, reviewers hoped to see more experiments incorporating different state-of-the-art techniques in domain adaptation, which can potentially further improve the performance. In summary, we have decided not to accept the work in its current state.

**Additional Comments On Reviewer Discussion:**

The Area Chair has carefully reviewed the paper and the discussions between the authors and reviewers and agrees that several concerns regarding the experiments remain unresolved. Please refer to the above for detailed comments.

---

### Decision · Program_Chairs · 2025-01-22

Reject